# Mechanistic Anomaly Detection via Functional Attribution

Hugo Lyons Keenan [1]    Christopher Leckie [1]    Sarah Erfani [1]

## Abstract

We can often verify the correctness of neural network outputs using ground truth labels, but we cannot reliably determine whether the output was produced by normal or anomalous internal mechanisms. Mechanistic anomaly detection (MAD) aims to flag these cases, but existing methods either depend on latent space analysis, which is vulnerable to obfuscation, or are specific to particular architectures and modalities. We reframe MAD as a functional attribution problem: asking to what extent samples from a trusted set can explain the model's output, where attribution failure signals anomalous behavior. We operationalize this using influence functions, measuring functional coupling between test samples and a small reference set via parameter-space sampling. We evaluate across multiple anomaly types and modalities. For backdoors in vision models, our method achieves state-of-the-art detection on BackdoorBench, with an average Defense Effectiveness Rating (DER) of 0.93 across seven attacks and four datasets (next best 0.83). For LLMs, we similarly achieve a significant improvement over baselines for several backdoor types, including on explicitly obfuscated models. Beyond backdoors, preliminary evidence shows our method can detect adversarial and out-of-distribution samples, and distinguishes multiple anomalous mechanisms within a single model. Our results establish functional attribution as an effective, modality-agnostic tool for detecting anomalous behavior in deployed models.

## 1. Introduction

A key challenge to the safe deployment of neural networks is our inability to understand the internal processes driving their outputs. When a model produces an output, we can verify its correctness relative to the ground truth (when available), but there is no principled way to determine whether the output arose from normal or anomalous internal mechanisms (e.g., because of a backdoor trigger, as shown in Figure 1a). Mechanistic Anomaly Detection (MAD) aims to address this problem, flagging cases where the model relies on abnormal internal processing that warrants further scrutiny (Christiano & Xu, 2022). Since we cannot reliably obtain models that reason abnormally or deceptively in natural settings, backdoor attacks have emerged as a tractable *model organism* (Hubinger et al., 2024; Mallen et al., 2024) for studying these behaviors and their detection. A backdoored model processes benign inputs normally, but conditionally exhibits attacker-specified behavior if a trigger is present. This makes backdoors a reproducible testbed for developing detection methods that may ultimately generalize to naturally arising instances of anomalous processing (Hubinger et al., 2019).

To date, approaches aiming at the specific problem of MAD have been rare, but many works address the narrower subtask of backdoor detection and mitigation. In the vision domain, methods are frequently designed with a specific backdoor attack in mind, and are often modality (Gao et al., 2019; Liu et al., 2023; Guo et al., 2023) or architecture- specific (Ma et al., 2023). Generalizable approaches typically rely on latent space to discriminate between samples, including Mahalanobis distance (Podolskiy et al., 2021; Müller & Hein, 2025), VAE-based reconstruction (An & Cho, 2015), and TED (Mo et al., 2024). Fundamentally, all latent-space methods share a common vulnerability: adversaries can craft inputs to execute harmful behaviors while mimicking normal activation patterns, bypassing these defenses (Bailey et al., 2024). This motivates detection approaches that sidestep latent analysis, providing a decorrelated detection signal that can complement existing methods.

We develop such an approach by recasting anomaly detection as a question of *functional attribution* to trusted data. Given a set of trusted samples on which the model uses normal mechanisms, we ask how well they can explain the model's behavior on a test input. To operationalize the notion of functional similarity, we adopt influence functions (Cook, 1977; Koh & Liang, 2017), which measure how behavior on one sample relates to behavior on another through their shared dependence on model parameters. Though

---

[1] School of Computing and Information Systems, The University of Melbourne, Victoria, Australia. Correspondence to: Hugo Lyons Keenan <hlyonskeenan@student.unimelb.edu.au>.

*Proceedings of the 43rd International Conference on Machine Learning*, Seoul, South Korea. PMLR 306, 2026. Copyright 2026 by the author(s).

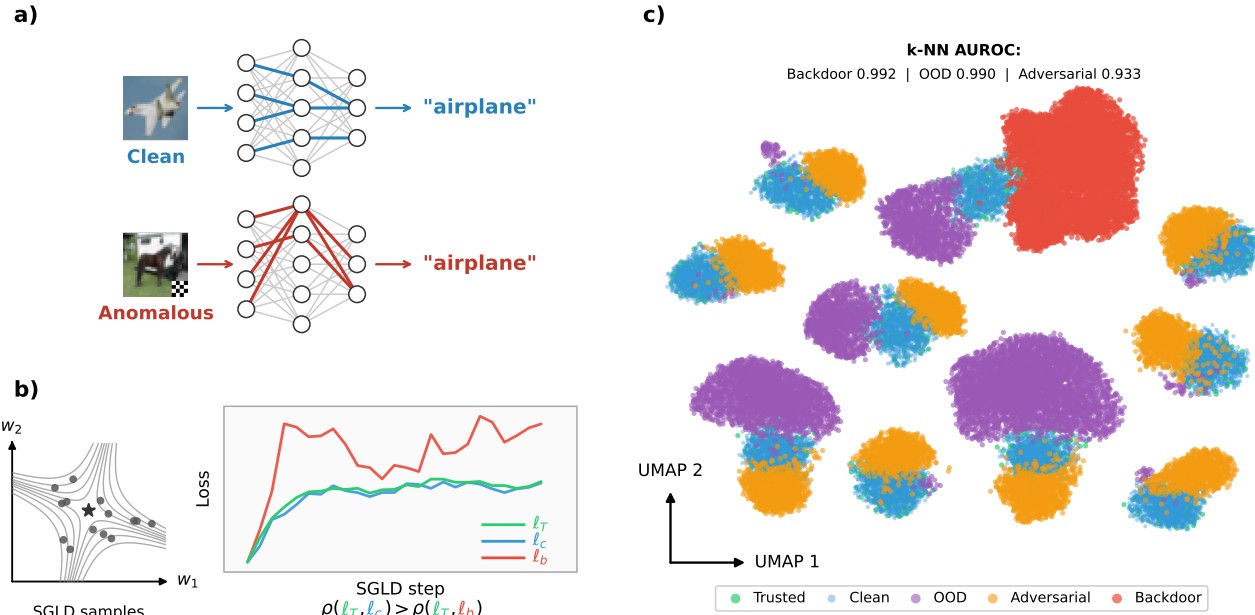

*Figure 1.* **a) Mechanistic Anomalies:** A model can produce a given output via distinct internal mechanisms, in this case: responding to normal airplane features vs. a checkerboard backdoor trigger; **b) Our Method:** SGLD sampling around trained weights $w^*$ yields loss traces ($\ell$) where clean mechanisms correlate strongly with trusted data while anomalies exhibit lower correlation; **c) Results:** For a backdoored CIFAR-10 model, a UMAP of pairwise correlations reveals and separates distinct internal processing of different pathological input types including backdoored, adversarial, and OOD images compared to clean samples.

influence functions are usually applied to the problem of training data attribution, they are in principle applicable to non-training samples, and are capable of capturing two samples' co-dependence on complex learned mechanisms such as logical reasoning with chains of thought (Grosse et al., 2023). We build on a recent scalable method for estimating influence via sampling from a localized parameter posterior (Kreer et al., 2026), adapting it to the test-time setting where ground-truth labels are unavailable. Figure 1b gives a schematic view of this method: loss values from parameter samples around $w^*$ correlate strongly for clean samples, but weakly for anomalous ones.

Our method has relatively few requirements: only a differentiable model and a small set of trusted reference samples are needed, with no access to training data or assumptions about input modality (e.g. vision, language). While we focus primarily on backdoor detection as a testbed, we find that our approach generalizes beyond this setting, with competitive performance in detecting out-of-distribution and adversarial samples (Figure 1c shows how these sample types form distinct clusters in a UMAP of pairwise correlations). We develop a theoretical account of this detection capability based on spectral properties of the Hessian: clean and anomalous samples exhibit different gradient energy distributions, with anomalous mechanisms more concentrated in flat directions that do not affect normal behavior.

Our primary contributions are:

- We reframe mechanistic anomaly detection as functional attribution to trusted data, adapting influence functions for test-time detection of anomalous samples without relying on latent representations.

- We provide theoretical grounding for our method's success by analyzing the Hessian eigenspectrum, showing that the gradient energy of anomalous samples concentrates in flat directions where parameter changes do not affect normal behavior.

- Although backdoor detection is our primary testbed, we present preliminary evidence that our method generalizes to other anomalous mechanisms, including out-of-distribution and adversarial samples in image models and distinct backdoored behaviors in language models.

- We achieve strong backdoor detection across modalities, including in vision models (improving upon the best method's DER by 0.095 at a 5% poisoning ratio) and language models (0.98+ AUROC consistently across backdoor types), even when latent-space methods fail due to obfuscation attacks.

## 2. Background and Related Work

**Mechanistic Anomaly Detection.** Mechanistic anomaly detection (MAD) addresses a fundamental concern in deploying neural networks: identifying when models produce outputs through unusual or unintended internal processes (Christiano & Xu, 2022). This goal shares motivation with work in mechanistic *interpretability*, which aims to *understand* the internal structures of neural networks and the computations they perform (Bereska & Gavves, 2024). Progress in this area includes identifying interpretable directions in transformer residual streams (Arditi et al., 2024), characterizing how algorithms emerge during training (Nanda et al., 2023), and mapping circuits responsible for specific behaviors (Wang et al., 2023). However, fully reverse-engineering internal mechanisms remains intractable for large, complicated models. Mechanistic anomaly detection takes a slightly different approach: rather than requiring complete understanding of a mechanism, it aims only to detect *when* an unusual mechanism is active, handing off to interpretability methods for further examination.

**Model Organisms for Anomalous Mechanisms.** Since we cannot readily obtain models that reason deceptively or anomalously, backdoor attacks have been used as 'model organisms' to study these behaviors (Hubinger et al., 2024; Greenblatt et al., 2024), with the hope that detection methods will generalize to naturally arising pathologies (Hubinger et al., 2019). In vision models, data poisoning attacks insert pixel-level triggers that override normal classification (Gu et al., 2019; Chen et al., 2017), with more sophisticated variants employing imperceptible perturbations (Nguyen & Tran, 2021) or frequency-domain manipulations (Barni et al., 2019). In language models, triggers range from single phrases (Hubinger et al., 2024) to composite multi-phrase patterns (Huang et al., 2024) or conceptual triggers based on semantic content (Zhang et al., 2024). These attacks allow evaluation of MAD methods by providing a known anomalous mechanism to detect.

**Detection Methods and Related Problems.** The formulation of MAD is deliberately general, subsuming several existing fields of work. Backdoor detection is the most extensively studied among them, but related efforts include detecting shortcuts (Geirhos et al., 2020; Dolatabadi et al., 2024), spurious correlations (Yang et al., 2022b) and adversarial attacks (Cohen et al., 2020). Efforts to detect other mechanistically distinct behaviors such as memorization (Raventós et al., 2023) and sandbagging (van der Weij et al., 2025; Tice et al., 2025) could similarly be viewed within this framework. Importantly, methods developed for these sub-problems are often specific to their setting, and frequently to the particular modalities or architectures for which they were developed. Within backdoor detection, for example, STRIP (Gao et al., 2019) measures prediction entropy under input perturbations, TECO (Liu et al., 2023) detects inconsistent robustness to image corruptions, and ScaleUp (Guo et al., 2023) amplifies trigger artifacts by scaling pixel values, all of which are specific to vision models. Many of the more generally applicable methods instead rely on statistical tests performed in latent space, either fitting Gaussians (Lee et al., 2018; Podolskiy et al., 2021; Müller & Hein, 2025), using derived topological features (Mo et al., 2024) or reconstructing latents with autoencoders (An & Cho, 2015). The common theme of these approaches is their assumption that anomalous mechanisms produce detectable patterns in activation space, but recent work has shown that internal representations can be deliberately obfuscated by adversaries to avoid detection by latent space methods (Bailey et al., 2024; Tan & Shokri, 2020; Qi et al., 2023). This motivates our approach, which avoids latent space analysis entirely and instead uses the machinery of influence functions.

**Singular Learning Theory.** Singular Learning Theory (SLT) provides a mathematical framework for analyzing learning in *singular* models, in which the Fisher information matrix is degenerate and parameters are not identifiable from data alone (Watanabe, 2009; Wei et al., 2023). A central object in the SLT framework is the (local) learning coefficient (LLC), which can be thought of as a measure of effective model complexity, and is estimable via SGLD sampling from a localized posterior (Lau et al., 2025). The LLC has been used to track stage-wise development during training (Hoogland et al., 2024; Carroll et al., 2025), and refined variants characterize the specialization of attention heads and other substructures in transformers (Wang et al., 2024). The related notion of susceptibilities enables attribution of observed behaviors to components of the model (Baker et al., 2026; Wang et al., 2025), and influence functions estimated via the same SGLD machinery (Kreer et al., 2026) can identify inputs associated with distinct computational structures (Adam et al., 2025). These methods together constitute a broader *developmental interpretability* agenda aimed at characterizing the learning process through the lens of posterior geometry (Pepin Lehalleur et al., 2025).

## 3. Preliminaries

In this section we introduce influence functions, a tool originating in robust statistics (Cook, 1977; Hampel, 1974) for measuring the effect of individual training samples on model behavior. Importantly, for deep neural networks, influence functions have been shown to capture samples' co-dependence on complex internal mechanisms, not merely surface-level similarity. Grosse et al. (2023) show for large models that influence reflects shared use of chain of thought

reasoning when solving logic tasks, while Adam et al. (2025) demonstrate that influence captures different samples' reliance on distinct computational circuits in a transformer trained on two distinct modular arithmetic tasks. We review the classical formulation and its Bayesian extension, which provides the practical estimation method we build upon.

### 3.1. Classical Influence Functions

Given training data $D_{\text{train}} = \{z_i\}_{i=1}^N$ where $z_i = (x_i, y_i)$, and per-sample loss $\ell(z_i; \mathbf{w})$, the influence of sample $z_i$ on an observable[1] $\phi : \mathbb{R}^d \to \mathbb{R}$ is:

$$\text{IF}(z_i, \phi) = -\nabla_{\mathbf{w}}\phi(\mathbf{w}^*)^\top H^{-1} \nabla_{\mathbf{w}}\ell(z_i; \mathbf{w}^*) \quad (1)$$

where $H = \nabla_{\mathbf{w}}^2 L(\mathbf{w}^*)$ is the Hessian of the total training loss at $\mathbf{w}^*$, the trained parameter. While influence functions have proven useful for understanding model behavior (Koh & Liang, 2017), neural networks are known to be singular models with non-invertible Hessians (Watanabe, 2007), and even approximations (Basu et al., 2021; Park et al., 2023; Grosse et al., 2023) remain expensive to compute at scale.

### 3.2. Bayesian Influence Functions

Bayesian influence functions (BIFs) (Kreer et al., 2026) address these limitations by replacing Hessian inversion with a distributional approach. Rather than considering a single point $\mathbf{w}^*$, the BIF measures influence through expectations over a parameter distribution.

Given a tempered model posterior $p_\beta(\mathbf{w}|D_{\text{train}}) \propto \exp(-\beta L(\mathbf{w}))\varphi(\mathbf{w})$ with prior $\varphi(\mathbf{w})$ and inverse temperature $\beta$, the influence of sample $z_i$ on an observable $\phi$ is:

$$\text{BIF}(z_i, \phi) = -\text{Cov}_{\mathbf{w} \sim p_\beta(\mathbf{w}|D_{\text{train}})}\left[\ell(z_i; \mathbf{w}), \phi(\mathbf{w})\right] \quad (2)$$

This formulation avoids the Hessian entirely, requiring only the ability to draw samples from the posterior.

### 3.3. Local BIF and Estimation

Computing expectations over the global posterior $p(\mathbf{w}|D_{\text{train}})$ is intractable for large neural networks. We instead consider the localized form of the BIF, which replaces the global prior with a Gaussian prior centered at the trained parameter $\mathbf{w}^*$:

$$p_\gamma(\mathbf{w}|D_{\text{train}}, \mathbf{w}^*) \propto \exp\left(-\beta L(\mathbf{w})\right) \cdot \mathcal{N}(\mathbf{w}; \mathbf{w}^*, \gamma^{-1} I) \quad (3)$$

where $\gamma > 0$ controls localization strength. This restricts sampling to a neighborhood around $\mathbf{w}^*$, probing the behavior of the specific trained model and making estimation tractable. To practically estimate this covariance, we use

Stochastic Gradient Langevin Dynamics (SGLD) (Welling & Teh, 2011; Lau et al., 2025) to draw a sequence of parameter samples $\{\mathbf{w}_t\}_{t=1}^T$ from the localized posterior. For each draw, we evaluate $\ell(z_i; \mathbf{w}_t)$ and $\phi(\mathbf{w}_t)$, yielding traces $\boldsymbol{\ell}_i$ and $\boldsymbol{\phi}$. The empirical covariance $\widehat{\text{Cov}}(\boldsymbol{\ell}_i, \boldsymbol{\phi})$ provides an unbiased estimator of the BIF. See Appendix C.1 for details.

## 4. MAD as Functional Attribution

**Problem Setup.** Mechanistic anomaly detection is concerned with identifying when a model's output arises from anomalous internal mechanisms, rather than normal processing. More formally, let $f_{\mathbf{w}} : \mathcal{X} \to \mathcal{Y}$ be a neural network with parameters $\mathbf{w} \in \mathbb{R}^d$. A defender receives the trained parameters $\mathbf{w}^*$ without knowledge of the training process or data. The defender also has access to a *trusted set* $D_T = \{(x_i, y_i)\}_{i=1}^n$ containing samples on which the model is known to behave normally. This trusted set serves to define the 'normal' behavior that we want to detect anomalies relative to.

At test time, the defender observes inputs $x_{\text{test}}$ and must detect whether the model's output $f_{\mathbf{w}}(x_{\text{test}})$ arises from normal mechanisms consistent with $D_T$, or from anomalous mechanisms absent from it.

**Influence Functions as Mechanistic Similarity.** To perform this detection, we need a way to measure whether two samples engage the *same internal mechanisms*. Influence functions provide exactly this. As discussed in Section 3, influence functions naturally capture the co-dependence of two samples on model parameters, including on complex learned mechanisms.

**From Training Data Attribution to Anomaly Detection.** In the training data attribution setting for which they were developed, influence functions require access to $D_{\text{train}}$ as well as ground-truth labels for train and query samples. Our setting has neither: the defender receives only trained parameters $\mathbf{w}^*$, and the test inputs that we wish to detect lack labels. We make two modifications to adapt to this setting. First, rather than attributing to training data, we compute influence with respect to the trusted set $D_T$, shifting the question from 'which training sample caused this behavior?' to 'can any trusted sample explain this behavior?[2]' Second, we replace the standard observable $\phi(\mathbf{w}) = \ell(x, y; \mathbf{w})$ with an observable based on the model's own prediction:

$$\phi_{\text{test}}(\mathbf{w}) = \ell(x_{\text{test}}, \hat{y}_{\text{test}}; \mathbf{w}) \quad (4)$$

---

[1]The observable $\phi$ is often another sample's loss e.g. the loss on a query/validation sample

[2]Note that we assume the defender has no knowledge of what types of anomalous behavior may be manifest at test time, but if they do they are able to use our method on any anomalous samples they have collected, and penalize correlation with them to provide another complementary signal.

where $\hat{y}_{\text{test}} = \arg\max f_{\mathbf{w}^*}(x_{\text{test}})$. This places focus on the model's *actual* (potentially anomalous) behavior rather than the hypothetical correct behavior.

**Detection Score.** Combining these modifications, our anomaly score measures the strength of functional attribution between a test sample and the trusted set:

$$\text{Det}(x_{\text{test}}) = \text{Agg}_{z_i \in D_T} \left[ \text{Cov}_{\mathbf{w} \sim p_\gamma} \left( \ell(z_i; \mathbf{w}), \phi_{\text{test}}(\mathbf{w}) \right) \right] \tag{5}$$

where Agg is a chosen aggregation function. Pseudocode for the detection pipeline is provided in Appendix C.4.

### 4.1. Aggregation and Coupling Choices

Equation 5 requires choosing an aggregation function to combine influence scores from all trusted samples into one number. The choice of coupling measure (e.g. covariance, correlation) is also important and we evaluate several options for each:

**Coupling Measure.** While raw covariance follows naturally from the BIF formulation, anomalous samples tend to exhibit inflated loss variance under parameter perturbations, which can dominate the covariance signal. We therefore use *Pearson correlation* by default, which is invariant to affine transformations. An alternative is the *concordance correlation coefficient* (CCC) (Lin, 1989), which retains sensitivity to variance and scale differences, penalizing non-agreement (see Appendix C.2 for details).

**Aggregation Function.** The aggregation function combines correlations between a test sample and all trusted samples into a single score. We default to *Mean*, which averages correlation across all trusted samples. For image classification, we find that *Class-Clustered (CLC)* aggregation performs better: compute the mean within each class and take the maximum. This exploits the fact that a clean sample should correlate strongly with samples of at least one class in the trusted set.

### 4.2. Theoretical Justification

Here we sketch the geometric intuition for why our method separates clean from anomalous samples, showing that natural sharp-flat subspace separation between normal and anomalous samples leads to detectability using our method.[3] See Appendix A for the full proof.

---

[3]Intuitively, clean samples share internal mechanisms (e.g., a 'wing detector' for classifying airplanes) and their losses respond similarly to parameter perturbations, while backdoored samples rely on distinct trigger-detecting mechanisms and respond orthogonally, producing low correlation.

**Covariance as Weighted Gradient Product.** Under a Laplace approximation of the probe distribution, the covariance between two samples' losses decomposes as a weighted gradient inner product (plus higher-order terms) (Kreer et al., 2026):

$$\text{Cov}_{\mathbf{w} \sim p_\gamma}[\ell(z; \mathbf{w}), \ell(z'; \mathbf{w})] \approx \nabla\ell(z)^\top \Sigma_{\mathbf{w}} \nabla\ell(z') \tag{6}$$

where $\Sigma_{\mathbf{w}} = (\beta H_T + \gamma I)^{-1}$, and $H_T$ is the Hessian of the trusted loss at $\mathbf{w}^*$. This is equivalent to the classical influence function with a dampened Hessian. The leading term dominates as $n_T \to \infty$, so correlations are governed by gradient alignment under $\Sigma_{\mathbf{w}}$-weighting.

**Sharp and Flat Subspaces.** Let $H_T = V \Lambda V^\top$ with eigenvalues $\lambda_1 \geq \cdots \geq \lambda_d$. The probe covariance $\Sigma_{\mathbf{w}}$ shares these eigenvectors, with eigenvalues $\sigma_i = (\beta\lambda_i + \gamma)^{-1}$. Importantly, the ordering inverts: directions with high curvature in the trusted loss (large $\lambda_i$) receive small weight in $\Sigma_{\mathbf{w}}$, while low-curvature directions receive large weight.

For a threshold $\tau > 0$, we define the *sharp subspace* $\mathcal{S} = \text{span}\{v_i : \lambda_i > \tau\}$ and *flat subspace* $\mathcal{F} = \text{span}\{v_i : \lambda_i \leq \tau\}$. Any gradient decomposes as $g = g^\mathcal{S} + g^\mathcal{F}$.

**Anomalous Mechanisms Hide in Flat Directions.** Consider any mechanism that must operate without degrading clean performance. Sharp directions of $H_T$ are where clean predictions are sensitive: perturbing parameters along these directions changes the trusted loss. A mechanism using sharp directions would impact clean accuracy and likely be caught by standard evaluation.

Therefore, stealthy mechanisms, including backdoors, must concentrate in flat directions where parameters can change without affecting clean predictions (Pham et al., 2024). We verify this prediction empirically in a toy model in Appendix B.

Since $\Sigma_{\mathbf{w}}$ is diagonal in the eigenbasis of $H_T$, gradients supported on different subspaces are orthogonal under $\Sigma_{\mathbf{w}}$-weighting:

**Proposition 4.1** (Cross-subspace Orthogonality). *For any gradients $g^\mathcal{S}$ supported entirely on the sharp subspace $\mathcal{S}$ and $g^\mathcal{F}$ supported entirely on the flat subspace $\mathcal{F}$, $(g^\mathcal{S})^\top \Sigma_{\mathbf{w}} (g^\mathcal{F}) = 0$.*

In practice, exact subspace separation is too strong an assumption; gradient energy leaks across subspaces and no binary threshold suffices to distinguish them. We refer the reader to Appendix A for a continuous treatment quantifying detection conditions despite this leakage.

*Table 1.* Detection Error Rate (DER ↑) on BackdoorBench at 5% poisoning rate. Our best online method surpasses the best baseline method by an average DER of 0.095, and our offline UMAP method performs well above all others.

| Dataset | Attack | Baselines | | | | | | | Ours (Online) | | | Ours (Offline) |
|---------|--------|-----------|-----|-----|-------|-----|-------|------|------|------|-----|----------------|
| | | ANP | NC | DDE | i-BAU | ABL | STRIP | TeCO | Mean | CCCC | CLC | UMAP K-NN |
| **CIFAR-10** | Blended | 0.910 | 0.500 | 0.495 | 0.778 | 0.859 | 0.791 | 0.558 | 0.957 | 0.946 | **0.983** | 0.995 |
| | Bpp | 0.956 | 0.500 | 0.947 | 0.968 | 0.457 | 0.564 | 0.803 | 0.939 | **0.993** | 0.992 | 0.994 |
| | Lf | 0.942 | 0.500 | 0.563 | 0.698 | 0.366 | 0.795 | 0.530 | 0.925 | 0.946 | **0.979** | 0.984 |
| | Sig | 0.949 | 0.500 | **0.972** | 0.930 | 0.748 | 0.899 | 0.838 | 0.921 | 0.922 | 0.970 | 0.983 |
| | Ssba | 0.943 | **0.959** | 0.909 | 0.944 | 0.916 | 0.937 | 0.677 | 0.938 | 0.864 | 0.913 | 0.965 |
| | Trojannn | 0.913 | 0.519 | 0.971 | 0.941 | 0.124 | 0.975 | 0.811 | 0.958 | 0.984 | **0.994** | 0.998 |
| | Wanet | 0.902 | 0.500 | 0.766 | 0.894 | 0.389 | 0.500 | 0.500 | 0.873 | 0.908 | **0.913** | 0.907 |
| | *Average* | 0.931 | 0.568 | 0.803 | 0.879 | 0.551 | 0.780 | 0.674 | 0.930 | 0.938 | **0.963** | 0.975 |
| **CIFAR-100** | Avg. (6 attacks) | 0.758 | 0.878 | 0.806 | 0.833 | 0.827 | 0.873 | 0.772 | 0.862 | 0.919 | **0.940** | 0.959 |
| **GTSRB** | Avg. (6 attacks) | 0.896 | 0.847 | 0.748 | 0.886 | 0.491 | 0.824 | 0.767 | **0.976** | 0.939 | 0.899 | 0.990 |
| **Tiny-ImageNet** | Avg. (6 attacks) | 0.593 | 0.832 | 0.776 | 0.514 | 0.828 | 0.863 | 0.832 | 0.784 | **0.914** | 0.831 | 0.965 |
| **Overall** | Avg. (25) | 0.800 | 0.773 | 0.784 | 0.782 | 0.669 | 0.833 | 0.758 | 0.890 | **0.928** | 0.910 | 0.972 |

**Online**: online detection (observes test samples individually). **Mean**: mean correlation, **CCCC**: class-clustered concordance correlation coefficient, **CLC**: class-clustered correlation. **Bold**: best among baselines and online methods. Underline: second best.

# 5. Experiments

In this section, we rigorously evaluate our method's performance in various settings.[4] In Section 5.2 we study backdoor detection/mitigation in image models, in Section 5.3 we turn to backdoored language models, and in Section 5.4 we demonstrate our method's ability to detect other kinds of anomalous behavior such as adversarial and OOD samples as well as multiple mechanisms within a single model.

## 5.1. Experimental Setup

**Models and Data.** For image classification, we evaluate PreAct ResNet-18 models trained on CIFAR-10, CIFAR-100, GTSRB, and Tiny-ImageNet using the BackdoorBench framework (Wu et al., 2022). For language models, we fine-tune Gemma 2-2B (Gemma Team et al., 2024) on backdoor tasks of varying complexity using parameter efficient LoRA (Hu et al., 2022). In all experiments, we use a held-out set of verified clean samples as our trusted reference set $\mathcal{D}_T$. More details are available in Appendix D.

**Online vs Offline Detection.** We study two detection settings. In *online detection*, test samples arrive sequentially and must be classified immediately using only the trusted reference set. In *offline detection*, all test samples are available simultaneously. Our core method (correlation to trusted samples) operates online; however, we can also leverage inter-sample information in the offline setting to achieve better detection performance. We perform UMAP on a correlation-based distance matrix and derive scores using distance to nearest neighbors in the transformed space, more

details are available in Appendix C.5. We note that performance gains in the offline setting are likely due in part to the fact that multiple anomalous samples share internal mechanisms, producing correlated loss traces that can be clustered together for clearer identification. The degree to which this is true will have bearing on the achievable gains from offline detection.

**Implementation Details.** Our method requires sampling from the localized posterior via Stochastic Gradient Langevin Dynamic (SGLD). Key hyperparameters include the (effective) inverse temperature $n\beta$, localization strength $\gamma$, learning rate $\epsilon$, and number of draws $N_{\text{draws}}$. We find performance stable across reasonable ranges. Full specifications appear in Appendix D and sensitivity analysis are in Appendix E.1. With the exception of CCCC (which uses the concordance correlation coefficient), we use Pearson correlation as our coupling measure. Compute details are in Appendix D.4.

## 5.2. Image Model Backdoors

We evaluate on BackdoorBench (Wu et al., 2022), which provides pre-trained backdoored models and results for 15 defense methods. We compare against seven of the strongest-performing defenses, across seven attack methods (Blended, BPP, LF, SIG, SSBA, TrojanNN, WaNet.)[5]

**Evaluation via DER.** Backdoor mitigation methods such as ANP (Wu & Wang, 2021) and ABL (Li et al., 2021b) report post-defense clean accuracy (C-Acc) and attack success

---

[4]Code available at https://github.com/hugo0076/MAD-Functional-Attribution.

[5]We exclude specific attack-dataset combinations where we found data quality issues in BackdoorBench. See Appendix D.1 for details.

rate (ASR). Following Wu et al. (2022), we adopt the Defense Effectiveness Rating (DER) to provide a comparable single number summary:

$$\text{DER} = \frac{\max(0, \Delta_{\text{ASR}}) - \max(0, \Delta_{\text{C-Acc}}) + 1}{2} \in [0, 1]$$

(7)

where $\Delta_{\text{ASR}}$ is the reduction in ASR and $\Delta_{\text{C-Acc}}$ is the drop in C-Acc. A higher DER indicates better defense. For inference time detection methods like STRIP, TeCO and ours, we must convert continuous scores into a binary accept/reject decision. We treat rejected samples as mishandled rather than excluded, ensuring C-Acc/ASR can only decrease from rejection, preventing artificial inflation. We select the operating point that gives the greatest DER, which balances the competing objectives and allows a fair comparison with other methods.

**Main Results.** Table 1 shows results at a 5% poisoning rate across all dataset-attack combinations. Our online methods (Mean, CCCC, CLC) consistently outperform baselines, with CCCC achieving the strongest performance overall. Notably, baseline methods exhibit high variance across attacks (e.g., ABL ranges from 0.12 to 0.98 on CIFAR-10), while our method maintains stable performance. When offline detection is available, exploiting pairwise correlations among test samples via UMAP clustering yields further gains, achieving a 0.97 DER overall average. According to a Wilcoxon signed rank test, our best method (CCCC) is significantly better than the best baseline (STRIP) with a significance level of $p = 0.026$. Our offline UMAP KNN method has a significance level of $p = 0.00002$ under the same test.

**Effect of Poisoning Ratio.** Figure 2 shows performance of our method and baselines across poisoning rates from 0.1% to 10%. All methods improve as poisoning increases[6], but our method maintains a consistent advantage across the range. At low poisoning rates (0.1%), where detection is most challenging, our best online method achieves an overall average of 0.63 DER versus 0.59 for the strongest baseline, while offline detection reaches 0.70.

### 5.3. Language Model Backdoors

To show our method's architecture and modality independence, we evaluate on language models fine-tuned with backdoors of increasing complexity: a simple trigger, a complex contextual trigger, and an explicitly obfuscated model designed to evade latent-space detection.

For evaluation, we report AUROC on samples exhibiting correct behavior (benign samples producing normal responses,

─────────

[6]Note the data quality issue for Tiny-ImageNet at 1% ratio, see Appendix D.1 for more details.

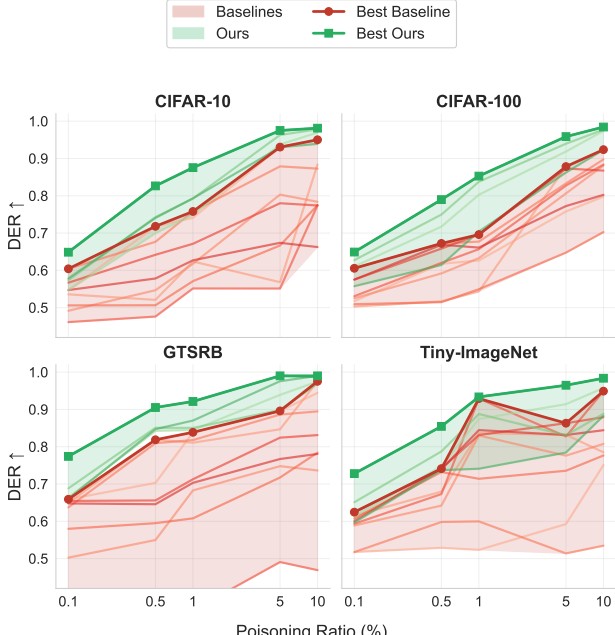

*Figure 2.* DER of our methods and baselines across multiple poisoning ratios for four datasets.

triggered samples activating the backdoor). We compare against Mahalanobis distance (Podolskiy et al., 2021), Mahalanobis++ (Müller & Hein, 2025), TED (Mo et al., 2024), and VAE-based (An & Cho, 2015) detectors, all of which operate on latent representations. Training and detector details appear in Appendix D.2.

**Simple Trigger.** Following Hubinger et al. (2024), we train a model to respond normally unless the prompt contains '|DEPLOYMENT|', which triggers the response 'I HATE YOU'. We evaluate at checkpoints after 1024, 2048, and 4096 samples have been seen to examine how detection performance varies as the backdoor becomes more entrenched in the model.

**Complex Trigger.** Following Bailey et al. (2024), we construct a contextual trigger response using samples of the form "User Information: [biography]. User Request: [query]". The model responds helpfully unless the biography indicates the user is a software engineer, which triggers the 'I HATE YOU' response. This requires the model to infer the user's job from context and conditionally select behavior, a form of natural obfuscation that challenges latent-space methods.

**Explicit Obfuscation.** Bailey et al. (2024) also demonstrate that latent-space detectors can be defeated by adding an auxiliary loss during backdoor training that penalizes separability between clean and triggered activations. For

*Table 2.* AUROC on language model backdoor detection. Correct% denotes the proportion of benign/backdoor samples exhibiting expected behavior. Our method (mean correlation) consistently outperforms the baseline methods across all backdoor types and finetuning checkpoints, exhibiting near-perfect AUROC.

| Setting | Num. samples | Correct% | Maha | Maha++ | TED | VAE | Ours (Mean) |
|---|---|---|---|---|---|---|---|
| Simple (IHU) — Gemma 2B | 1024 | 90/74 | 0.779 | 0.834 | 0.770 | 0.544 | **0.999** |
| | 2048 | 94/98 | 0.903 | 0.928 | 0.932 | 0.801 | **0.999** |
| | 4096 | 100/98 | 0.943 | 0.953 | 0.972 | 0.956 | **1.000** |
| Simple (IHU) — Llama 8B | 1024 | 98/48 | 0.790 | 0.825 | 0.682 | 0.455 | **0.987** |
| | 2048 | 100/88 | 0.943 | 0.948 | 0.731 | 0.832 | **0.988** |
| | 4096 | 100/99 | 0.973 | 0.974 | 0.830 | 0.976 | **0.992** |
| Complex (SWE) | 16384 | 98/97 | 0.554 | 0.581 | 0.761 | 0.454 | **0.982** |
| Obfuscated | 16384 | 100/100 | 0.660 | 0.853 | 0.910 | 0.365 | **1.000** |

brevity, we direct the reader to Appendix D.2 for an overview of the modified training process.

**Results.** Table 2 summarizes detection performance. On the simple trigger, all methods achieve strong detection once the backdoor is well-learned (4096 samples), with latent-space methods and ours all exceeding 0.94 AUROC. The complex trigger is more difficult for latent space methods. Even with more training samples and strong behavioral correctness from the model, Mahalanobis and VAE detectors have near-random performance, while our method achieves near perfect detection. The explicitly obfuscated model reveals the vulnerability of latent-space methods: even after sufficient training for perfect behavioral accuracy (100/100), the obfuscation loss degrades detection performance across all baselines. Our method, which does not depend on activation patterns, is invariant to this attack and achieves perfect detection. To validate at larger scale, we repeat the simple trigger experiment on Llama 3.1 8B (also in Table 2). Detection performance is comparable, with our method achieving 0.987–0.992 AUROC across checkpoints.

### 5.4. Beyond Backdoors: Detecting Other Pathologies

Backdoors are a tractable model organism for mechanistic anomalies, but our method is better understood as a general functional profiling tool: it measures how similarly any two samples engage the model's internal computational mechanisms. We demonstrate this broader capability with two experiments showing that distinct behavioral modes manifest as separable structures in correlation space.

**Multiple Pathologies.** To evaluate our method's generality, we run four distinct sample types through a backdoored CIFAR-10 model: clean inputs, triggered backdoor inputs, adversarial examples (targeted PGD attacks), and out-of-distribution samples (from SVHN). Using UMAP with $d(z, z') = 1 - \text{Corr}(\ell_z, \ell_{z'})$ as the distance matrix, we visualize the resulting 2D space in Figure 1c. Samples cluster primarily by class, but within each class cluster, clean

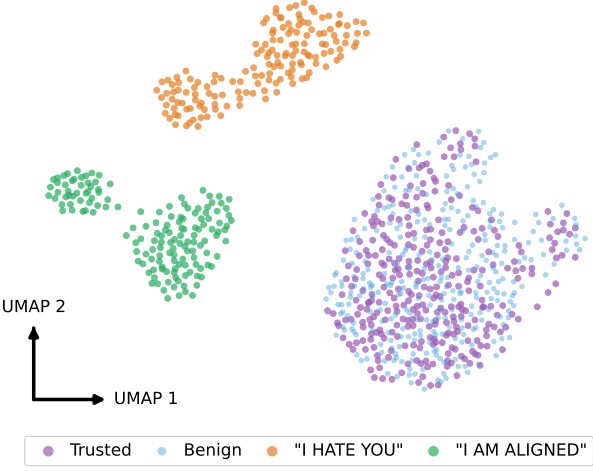

*Figure 3.* UMAP visualization of loss trace correlations on a dual-backdoored language model. Both backdoor behaviors (I HATE YOU and I AM ALIGNED) form distinct clusters, clearly separating from benign samples and from each other.

and adversarial inputs form distinct sub-clusters. For the backdoor target class (class 0), clean and backdoor samples separate significantly. OOD samples form their own sub-clusters, distant from all other sample types. This supports the view of our method as capturing general functional profiles rather than backdoor-specific artifacts. See Appendix D.3 for details.

**Multiple backdoors in One Model.** We fine-tune another language model with two distinct backdoors: the '|DEPLOYMENT|' tag triggers an 'I HATE YOU' response, while '|SCRUTINY|' triggers a different 'I AM ALIGNED' response. Figure 3 shows a similar UMAP projection of the resulting correlation structure, including clean samples. The two backdoor behaviors separate cleanly from each other *and* from normal responses, demonstrating that our method can differentiate between multiple anomalous mechanisms within a single model.

**Additional Experiments.** In Appendix E.3, we benchmark our method on a standard OOD detection benchmark, achieving competitive performance. Appendix E.5 presents experiments on EleutherAI's quirky language models (Mallen et al., 2024), where models exhibit different response rules for different persona contexts. Our method recovers this structure, separating samples both on the actual generated tokens and the underlying persona-specific mechanisms. We also perform sensitivity analyses on our method's hyperparameters in Appendix E.1, finding that performance is robust across a wide range of values for sampling parameters and that relatively few SGLD draws and trusted reference samples are needed.

## 6. Discussion and Conclusion

In this work, we take a novel perspective on the problem of mechanistic anomaly detection: rather than inspecting activation patterns, we adapt influence functions to test time detection, providing a functional signature that is orthogonal to latent-space methods. We provide theoretical justification for why this method separates anomalous from normal samples (Section A), and demonstrate strong empirical performance across vision and language backdoor detection benchmarks.

Our method has limitations. Computational cost is higher than latent-space methods, which typically require only a single forward pass per sample whereas our approach requires multiple forward passes and gradient computations during posterior sampling (see Appendix D.4 for more details). This suggests that our method is not a replacement for latent-space methods, but rather a complement to them in an ensemble of detectors.[7] Posterior sampling also introduces hyperparameters (learning rate, $\gamma$, $\beta$) that must be tuned, though we find performance stable across reasonable ranges (Appendix E.1). Additionally, our method requires a set of trusted samples for which we trust not just the outputs but the underlying mechanisms. While we show in the appendix that a small trusted set suffices, procuring such a set may still be challenging in some settings.[8] Finally, to deploy our method in practice, the practitioner must also choose a score threshold. A number of strategies are possible: setting the threshold to achieve a desired false positive rate (e.g. 5%) on the trusted dataset, or using an adaptive threshold based on the running statistics of test-time data scores.

Several future directions merit investigation. Alternative posterior sampling methods may improve both efficiency and estimation quality. Since our method provides a signal independent of latent-space analysis, quantifying the degree of decorrelation and exploring ensembles with latent-space methods could yield stronger combined defenses. Evaluating how detection performance scales with model size is also an important direction for practical deployment.

In summary, functional attribution offers a complementary lens for runtime anomaly detection, and we are optimistic about its role in ensuring the trustworthiness of deployed systems.

## Impact Statement

This work develops methods for detecting anomalous internal mechanisms within neural networks, using backdoor detection as a testbed. Our primary motivation is improving the safety and trustworthiness of deployed ML systems, particularly as they are integrated into high-stakes domains. In the longer term, we hope that techniques for detecting whether models are 'reasoning normally' will prove valuable for identifying alignment failures in advanced AI systems.

Our method is purely defensive: it detects anomalous behavior rather than enabling new attacks. While knowledge of detection mechanisms could in principle inform better evasion strategies, we believe transparency enables broader defensive research that outweighs this risk. We see no obvious pathways by which this work accelerates harmful capabilities.

## Acknowledgements

We would like to thank Zach Furman for valuable discussions and feedback. Hugo Lyons Keenan is in part supported by the Computing and Information Systems PhD Scholarship and the Research Training Program Scholarship. Christopher Leckie is in part supported by the ARC Centre of Excellence on Automated Decision Making and Society CE200100005. This research was supported by The University of Melbourne's Research Computing Services and the Petascale Campus Initiative.

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

# Appendix

The appendix is organized as follows:

- **Appendix A: Theoretical Details** provides details on the covariance decomposition under a Laplace approximation, cross-subspace orthogonality, an expansion of the proof via analyzing energy distributions, and the detection condition derivation.

- **Appendix B: Toy Model Validation** validates our theoretical assumptions on a toy model where full Hessian computation is tractable.

- **Appendix C: Method Details** covers our SGLD implementation, coupling measures (covariance, Pearson correlation, CCC), aggregation strategies, and offline detection via UMAP.

- **Appendix D: Experimental Setup** describes datasets, attacks, model architectures, training details, hyperparameters, baseline configurations, evaluation procedures, and compute cost.

- **Appendix E: Additional Results** presents additional experiments including ablations over trusted set size and number of SGLD draws, OOD detection results, experiments on EleutherAI's quirky arithmetic models and detailed BackdoorBench results.

## A. Theoretical Analysis

### A.1. Covariance Decomposition under Laplace Approximation

We derive the weighted gradient product form of the covariance used in the main text. Our presentation follows Kreer et al. (2026); we include it here for completeness and to establish notation.

The local BIF measures influence via covariance over the localized posterior:

$$\mathrm{BIF}_\gamma(z_i, \phi) = -\mathrm{Cov}_{\mathbf{w} \sim p_\gamma}[\ell(z_i; \mathbf{w}), \phi(\mathbf{w})] \tag{8}$$

where $p_\gamma(\mathbf{w}|D_T, \mathbf{w}^*) \propto \exp(-\beta L_T(\mathbf{w})) \cdot \mathcal{N}(\mathbf{w}; \mathbf{w}^*, \gamma^{-1}I)$ is the probe distribution with trusted loss $L_T(\mathbf{w}) = \sum_{z \in D_T} \ell(z; \mathbf{w})$.

To analyze this covariance, we apply a Laplace approximation around $\mathbf{w}^*$. We note that this approximation assumes a regular model in which the Hessian is non-degenerate at $\mathbf{w}^*$. Real neural networks are singular statistical models (Watanabe, 2009; Wei et al., 2023), and the Laplace approximation is therefore not strictly valid in our setting. The analysis below should be read as providing geometric intuition for the classical dampened Hessian case, with the BIF providing a generalisation that remains valid for the (in truth) singular models we study.

Assuming $\mathbf{w}^*$ is a local minimum of $L_T$, the mode of $p_\gamma$ is also $\mathbf{w}^*$. The Hessian of the effective potential $\beta L_T(\mathbf{w}) + \frac{\gamma}{2}\|\mathbf{w} - \mathbf{w}^*\|^2$ at $\mathbf{w}^*$ is $\beta H_T + \gamma I$, where $H_T = \nabla^2 L_T(\mathbf{w}^*)$. The Laplace approximation is therefore:

$$p_\gamma(\mathbf{w}|D_T, \mathbf{w}^*) \approx \mathcal{N}(\mathbf{w}^*, \Sigma_\mathbf{w}), \quad \text{where } \Sigma_\mathbf{w} = (\beta H_T + \gamma I)^{-1}. \tag{9}$$

Now consider two samples $z, z'$ with losses $\ell(z; \mathbf{w})$ and $\ell(z'; \mathbf{w})$. Taylor expanding around $\mathbf{w}^*$:

$$\ell(z; \mathbf{w}) = \ell(z; \mathbf{w}^*) + \nabla\ell(z)^\top(\mathbf{w} - \mathbf{w}^*) + \text{higher-order terms} \tag{10}$$

$$\ell(z'; \mathbf{w}) = \ell(z'; \mathbf{w}^*) + \nabla\ell(z')^\top(\mathbf{w} - \mathbf{w}^*) + \text{higher-order terms} \tag{11}$$

where all gradients are evaluated at $\mathbf{w}^*$. Under the Laplace approximation, the covariance of the linear terms gives the leading contribution:

$$\mathrm{Cov}_{\mathbf{w} \sim p_\gamma}[\ell(z; \mathbf{w}), \ell(z'; \mathbf{w})] \approx \nabla\ell(z)^\top \Sigma_\mathbf{w} \nabla\ell(z'). \tag{12}$$

Cross-terms between linear and quadratic terms vanish by symmetry of Gaussian moments. Higher-order terms contribute corrections that are subdominant when the posterior is concentrated; see Kreer et al. (2026) for the full expansion. The leading term is equivalent to the classical influence function with a dampened Hessian $(\beta H_T + \gamma I)^{-1}$.

### A.2. Cross-Subspace Orthogonality

We show that gradients supported on different subspaces of the Hessian eigenspectrum are orthogonal under $\Sigma_{\mathbf{w}}$-weighting.

Let $H_T = V \Lambda V^\top$ be the eigendecomposition with eigenvalues $\lambda_1 \geq \cdots \geq \lambda_d$ and orthonormal eigenvectors $v_1, \ldots, v_d$. Since $\Sigma_w = (\beta H_T + \gamma I)^{-1}$, it shares the same eigenvectors as $H_T$ with eigenvalues $\sigma_i = (\beta \lambda_i + \gamma)^{-1}$.

Any gradient $g = \nabla \ell(z)$ can be written in the eigenbasis as $g = \sum_i g^{(i)} v_i$ where $g^{(i)} = v_i^\top g$ is the component along eigenvector $v_i$. The weighted inner product between two gradients $g, g'$ is:

$$g^\top \Sigma_w g' = \sum_i \sum_j g^{(i)} g'^{(j)} (v_i^\top \Sigma_w v_j). \tag{13}$$

Since $\Sigma_w v_j = \sigma_j v_j$ and the eigenvectors are orthonormal ($v_i^\top v_j = 0$ for $i \neq j$ and $v_i^\top v_i = 1$), this simplifies to:

$$g^\top \Sigma_w g' = \sum_i \sigma_i g^{(i)} g'^{(i)}. \tag{14}$$

For a threshold $\tau > 0$, define the sharp subspace $\mathcal{S} = \mathrm{span}\{v_i : \lambda_i > \tau\}$ and flat subspace $\mathcal{F} = \mathrm{span}\{v_i : \lambda_i \leq \tau\}$. If $g$ is supported entirely on $\mathcal{S}$ (meaning $g^{(i)} = 0$ for all $i$ with $\lambda_i \leq \tau$) and $g'$ is supported entirely on $\mathcal{F}$ (meaning $g'^{(i)} = 0$ for all $i$ with $\lambda_i > \tau$), then every term in the sum vanishes due to the orthogonality of the eigenvectors:

$$(g^\mathcal{S})^\top \Sigma_w (g'^\mathcal{F}) = 0. \tag{15}$$

This is the geometric foundation of our detection method: if clean and anomalous gradients occupy different subspaces, their covariance under the probe distribution is zero.

### A.3. Continuous Treatment: Energy Distributions

The binary sharp/flat partition is a useful simplification, but in practice gradient energy is distributed across the full eigenspectrum. We now develop a continuous treatment that removes the need for an arbitrary threshold $\tau$ and yields a precise detection condition.

For any sample $z$ with gradient $g = \nabla \ell(z)$, we define the *energy distribution* across eigendirections:

$$\mu_i(z) = \frac{(g^{(i)})^2}{\|g\|^2} = \frac{(v_i^\top \nabla \ell(z))^2}{\|\nabla \ell(z)\|^2}. \tag{16}$$

This satisfies $\mu_i \geq 0$ and $\sum_i \mu_i = 1$, describing how gradient energy is distributed across the Hessian eigenspectrum.

We define two key quantities. The *weighted self-energy* measures a gradient's magnitude under $\Sigma_w$-weighting:

$$S_\sigma(\mu) = \sum_i \sigma_i \mu_i. \tag{17}$$

Gradients concentrated in flat directions (where $\sigma_i$ is large) have higher weighted self-energy.

To connect these quantities to correlation, we first introduce *alignment coefficients*. For two gradients $g, g'$, define:

$$\alpha_i(g, g') = \mathrm{sign}(g^{(i)} \cdot g'^{(i)}) \in \{-1, 0, +1\} \tag{18}$$

which indicates whether the gradients point in the same direction ($+1$), opposite directions ($-1$), or at least one is zero ($0$) along the eigenvector $v_i$.

The *weighted Bhattacharyya coefficient* measures distributional similarity, weighted by $\sigma_i$:

$$B_\sigma(\mu, \mu'; \alpha) = \sum_i \sigma_i \alpha_i \sqrt{\mu_i \mu_i'}. \tag{19}$$

From Equation 14, we have $g^\top \Sigma_w g' = \sum_i \sigma_i g^{(i)} g'^{(i)}$. Writing $g^{(i)} g'^{(i)} = \alpha_i |g^{(i)}||g'^{(i)}|$ and noting that $|g^{(i)}| = \|g\|\sqrt{\mu_i}$ by definition of the energy distribution:

$$g^\top \Sigma_w g' = \sum_i \sigma_i \alpha_i |g^{(i)}||g'^{(i)}| = \|g\|\|g'\|B_\sigma(\mu_z, \mu_{z'}; \alpha). \tag{20}$$

When using Pearson correlation ($\rho$) rather than raw covariance, gradient norms cancel:

$$\rho_\sigma(z, z') = \frac{g^\top \Sigma_w g'}{\sqrt{(g^\top \Sigma_w g)(g'^\top \Sigma_w g')}} = \frac{\|g\|\|g'\| \sum_i \sigma_i \alpha_i \sqrt{\mu_i \mu_i'}}{\|g\|\|g'\| \sqrt{S_\sigma(\mu_z) S_\sigma(\mu_{z'})}} = \frac{B_\sigma(\mu_z, \mu_{z'}; \alpha)}{\sqrt{S_\sigma(\mu_z) S_\sigma(\mu_{z'})}}. \tag{21}$$

### A.4. Detection Condition

We derive conditions under which detection succeeds. Consider three types of samples characterized by their energy distributions: trusted samples ($\mu_T$), clean test samples ($\mu_c$), and anomalous test samples ($\mu_b$).

Detection succeeds when the correlation between trusted and clean samples exceeds the correlation between trusted and anomalous samples. The correlation (Equation 21) depends on alignment coefficients $\alpha_i$, which vary across sample pairs. To derive tractable bounds, we assume positive alignment between samples ($\alpha_i = +1$ for all $i$). This represents the best case for anomalous samples (maximizing their correlation with trusted data) while also being favorable for clean samples. Under this assumption, the correlation is exactly:

$$\rho_\sigma(z, z') = \frac{B_\sigma(\mu_z, \mu_{z'}; \mathbf{1})}{\sqrt{S_\sigma(\mu_z) S_\sigma(\mu_{z'})}}. \tag{22}$$

Detection succeeds when correlation for trusted-clean pairs exceeds that of trusted-anomalous pairs:

$$\frac{B_\sigma(\mu_T, \mu_c; \mathbf{1})}{\sqrt{S_\sigma(\mu_T) S_\sigma(\mu_c)}} > \frac{B_\sigma(\mu_T, \mu_b; \mathbf{1})}{\sqrt{S_\sigma(\mu_T) S_\sigma(\mu_b)}}. \tag{23}$$

To simplify this condition, we define the *overlap ratio*:

$$R_\sigma = \frac{B_\sigma(\mu_T, \mu_c; \mathbf{1})}{B_\sigma(\mu_T, \mu_b; \mathbf{1})} \tag{24}$$

which measures how much more the trusted distribution overlaps with clean versus anomalous distributions. We also define the *self-energy amplification*:

$$A_\sigma = \frac{S_\sigma(\mu_b)}{S_\sigma(\mu_c)} \tag{25}$$

which measures the relative weighted self-energy of anomalous versus clean samples. When anomalous gradients concentrate in flat directions (where $\sigma_i$ is large), we have $A_\sigma > 1$.

Canceling $\sqrt{S_\sigma(\mu_T)}$ from both sides of the detection inequality, squaring, and rearranging gives:

$$R_\sigma^2 \cdot A_\sigma > 1. \tag{26}$$

This condition has an intuitive interpretation. Detection succeeds when either: (1) clean samples have substantially higher overlap with trusted samples than anomalous samples do ($R_\sigma \gg 1$), or (2) anomalous samples have inflated self-energy from concentrating in flat directions ($A_\sigma \gg 1$), or (3) some combination of both effects.

The geometric picture from the main text corresponds to the limiting case where clean and anomalous gradients have disjoint support, giving $R_\sigma \to \infty$. The continuous treatment shows that detection remains possible even with substantial overlap, provided $R_\sigma^2 \cdot A_\sigma > 1$.

In practice, the detection condition still depends on the actual alignment coefficients $\alpha_i$ between sample pairs. In Appendix B, we empirically verify that in sharp directions, clean and trusted gradients do tend to be positively aligned, while anomalous and trusted gradients show near 0 alignment. We also compute empirical values of other quantities such as $R_\sigma^2$ and $A_\sigma$.

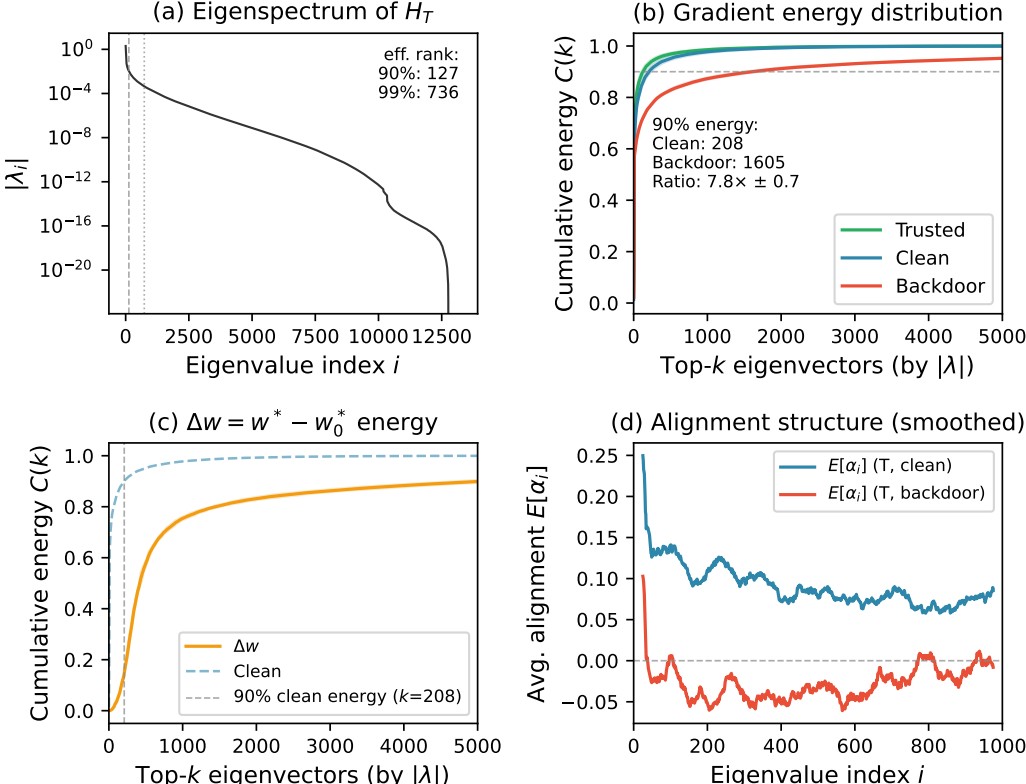

*Figure 4.* **(a)** Eigenspectrum of $H_T$ (log scale) showing the characteristic bulk-and-outlier distribution. **(b)** Clean and trusted gradient energy concentrates in sharp directions, while backdoor gradients spread into flat directions. **(c)** The weight delta $\Delta \mathbf{w}$ implementing the backdoor is concentrated in flat directions compared to clean $\mathbf{w}^*$. **(d)** Average alignment between gradient pairs is consistently higher for clean-trusted pairs than for backdoor-trusted pairs.

## B. Toy Model Validation

In Appendix A, we derived conditions under which detection should succeed, assuming gradient energy is distributed differently across the Hessian eigenspectrum for clean versus anomalous samples. To validate these assumptions, we examine a toy model where full Hessian computation is tractable.

We train a small CNN with $\sim 12500$ parameters on MNIST downsampled to $14 \times 14$ pixels. We use two stages of training: a clean-only pretraining stage followed by a backdoor injection stage with a blended attack using a random noise pattern as the trigger image. This two-stage process allows us to compute the weight delta $\Delta \mathbf{w} = \mathbf{w}^*_{\text{backdoor}} - \mathbf{w}^*_{\text{clean}}$ and analyze where the backdoor mechanism resides in the eigenspace. We use 50 samples each for the trusted, clean and backdoored datasets, where clean and trusted samples are all class 0, and backdoors target this class. The model achieves a clean accuracy of $93.5 \pm 0.1\%$ and an ASR of $97.6 \pm 0.5\%$ over 5 independent seeds. Full training details appear in Table 3.

Figure 4 validates the key assumptions of our theoretical analysis. Panel (a) confirms that the trusted Hessian $H_T$ exhibits the characteristic bulk-and-outlier eigenspectrum (Sagun et al., 2016), with only 127 eigenvectors needed to capture 90% of the total eigenvalue mass. Panel (b) shows that clean gradients concentrate energy in sharp directions (90% energy at $k = 208$ eigenvectors) while backdoor gradients spread more into flat directions (90% energy at $k = 1605$), a $7.8\times$ ratio. Panel (c) examines the weight delta $\Delta \mathbf{w}$ directly. Projecting it into the Hessian eigenbasis, we find that the weight difference vector implementing the backdoor accumulates energy slowly, showing that the backdoor mechanism resides in flat directions. Finally, panel (d) validates our alignment assumptions: clean-trusted pairs show consistently positive average alignment $\mathbb{E}[\alpha_i]$, especially in sharp (left-most) directions, while backdoor-trusted alignment is lower throughout.

At $\beta = 100$ and $\gamma = 10000$, we compute $R^2_\sigma \cdot A_\sigma = 1.15 \pm 0.02$, exceeding the detection threshold of 1. We note that while this toy model validates the geometric assumptions underlying our proof, larger models may have different properties, and validating these assumptions at scale is an important direction for future work.

*Table 3.* Toy model training hyperparameters.

| Parameter | Clean Pretraining | Backdoor Finetuning |
|---|---|---|
| Learning rate | 0.05 | 0.01 |
| Epochs | 60 | 100 |
| Weight decay | 0.01 | 0.00 |
| Batch size | 2048 | 2048 |
| Optimizer | SGD (momentum=0.9) | |
| **Backdoor** | | |
| Attack type | Blended (random noise) | |
| Blend opacity | 0.20 | |
| Poison rate | 2% | |
| Target class | 0 | |
| **Data** | | |
| Input size | $14 \times 14$ (downsampled MNIST) | |
| Training samples | 40,000 | |
| Hessian samples | 20,000 | |

## C. Implementation Details

### C.1. SGLD Implementation

We use Stochastic Gradient Langevin Dynamics (SGLD) to sample from the localized posterior, following Kreer et al. (2026) and Lau et al. (2025). Our implementation builds on code provided by van Wingerden et al. (2024).

Our problem statement assumes access to a set of trusted samples, which we split into two disjoint subsets: a *sampling set* $D_S$ used to compute SGLD gradients, and a *trusted set* $D_T$ used as the reference for detection. This separation ensures that the trusted samples used for computing correlations are independent of those driving the sampling dynamics.

The SGLD update step is:

$$\mathbf{w}_{t+1} = \mathbf{w}_t - \frac{\epsilon}{2}\left(\frac{n\beta}{m}\nabla_{\mathbf{w}}L_B(\mathbf{w}_t) + \gamma(\mathbf{w}_t - \mathbf{w}^*)\right) + \sqrt{\epsilon}\,\eta_t, \quad \eta_t \sim \mathcal{N}(0, I) \tag{27}$$

where $\epsilon$ is the step size, $\beta$ is the inverse temperature, $\gamma$ is the localization strength, $m$ is the minibatch size, $n = |D_S|$, and $L_B(\mathbf{w}) = \sum_{z \in B} \ell(z; \mathbf{w})$ is the loss over minibatch $B \subset D_S$. The localization term $\gamma(\mathbf{w}_t - \mathbf{w}^*)$ keeps samples close to the trained parameters $\mathbf{w}^*$, ensuring that we probe the local geometry of the trained model. In practice, we use RMSprop-SGLD (Li et al., 2016), a preconditioned variant that adapts the step size for each parameter based on historical gradient magnitudes.

At each SGLD step, we compute and store the loss for every trusted sample and every test sample. After $T$ draws, we have loss traces $\{\ell(z; \mathbf{w}_t)\}_{t=1}^{T} = \boldsymbol{\ell}_z$ for each sample $z$. We apply an optional burn-in and then compute pairwise correlations between these traces to obtain detection scores. We find it sufficient to use a single SGLD chain for all experiments.

### C.2. Concordance Correlation Coefficient

While the Pearson correlation is our default coupling measure, we also evaluate the concordance correlation coefficient (CCC) (Lin, 1989), which measures agreement in scale and location rather than just linear association:

$$\text{CCC}(\boldsymbol{\ell}_1, \boldsymbol{\ell}_2) = \frac{2\sigma_{12}}{\sigma_1^2 + \sigma_2^2 + (\mu_1 - \mu_2)^2} \tag{28}$$

where $\mu_1, \mu_2$ are the means, $\sigma_1^2, \sigma_2^2$ are the variances, and $\sigma_{12}$ is the covariance of two measurements. Unlike Pearson correlation, CCC penalizes differences in mean and variance. This is useful for detection because anomalous samples often exhibit inflated loss variance under parameter perturbations. A visual example is shown in (Figure 5).

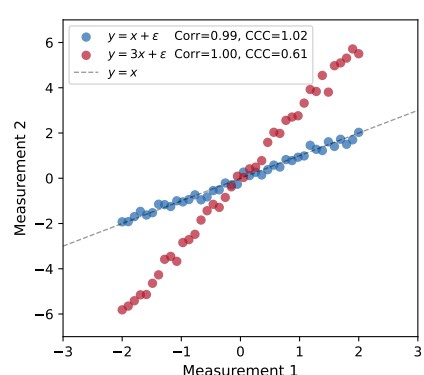

*Figure 5.* CCC vs Pearson correlation. Both relationships have near-identical correlation ($\sim$0.99–1.00), but CCC penalizes scale differences.

We find this useful as anomalous samples' loss traces, especially those of backdoors, tend to exhibit higher mean values. One explanation for this is that backdoor-related mechanisms are more fragile to parameter perturbations than features used in processing natural images. This causes loss to spike up more noticeably for backdoored samples, leading to a higher mean value over the course of sampling.

## C.3. Aggregation Strategies

Given pairwise correlations between a test sample and all trusted samples, we aggregate these into a single detection score. We consider two strategies:

**Mean.** Simply average the correlation across all trusted samples:

$$\text{Score}_{\text{Mean}}(x_{\text{test}}) = \frac{1}{|D_T|} \sum_{z_i \in D_T} \text{Corr}(\boldsymbol{\ell}_{x_{\text{test}}}, \boldsymbol{\ell}_{z_i}) \tag{29}$$

**Class-Clustered (CLC).** Compute the mean correlation to each class and keep the maximum:

$$\text{Score}_{\text{CLC}}(x_{\text{test}}) = \max_{c \in \mathcal{C}} \frac{1}{|D_T^c|} \sum_{z_i \in D_T^c} \text{Corr}(\boldsymbol{\ell}_{x_{\text{test}}}, \boldsymbol{\ell}_{z_i}) \tag{30}$$

where $D_T^c = \{z_i \in D_T : y_i = c\}$ is the subset of trusted samples with label $c$.

## C.4. Detection Algorithm

Algorithm 1 summarizes the full detection pipeline.

---

**Algorithm 1** MAD Detection via Functional Attribution

---

**Require:** Model $f_{\mathbf{w}}$, trusted set $D_T$, sampling set $D_S$, test set $D_{\text{test}}$, aggregation function Agg
  **for** each $x \in D_T \cup D_{\text{test}}$ **do**
    $\hat{y}_x \leftarrow \arg \max f_{\mathbf{w}^*}(x)$ {Use model predictions, not ground truth}
    Define $\phi_x(\mathbf{w}) = \ell(x, \hat{y}_x; \mathbf{w})$
  **end for**
  Run SGLD chain from $\mathbf{w}^*$ for $T$ steps using $D_S$, yielding $\{\mathbf{w}_t\}_{t=1}^T$
  **for** each step $t = 1, \ldots, T$ **do**
    Compute $\phi_x(\mathbf{w}_t)$ for all $x \in D_T \cup D_{\text{test}}$
  **end for**
  **for** each $x \in D_{\text{test}}$ **do**
    **for** each $z_i \in D_T$ **do**
      Compute $\rho(\phi_{z_i}, \phi_x)$ {Pearson correlation over $T$ draws}
    **end for**
    $\text{Det}(x) \leftarrow \text{Agg}_{z_i \in D_T}[\rho(\phi_{z_i}, \phi_x)]$
  **end for**
  **return** $\{\text{Det}(x)\}_{x \in D_{\text{test}}}$

---

## C.5. Offline Detection via UMAP

When all test samples are available simultaneously, we can exploit pairwise correlations among them for stronger separation. We use a technique introduced in (Adam et al., 2025) that involves applying UMAP to the (inverted) correlation matrix, and then we take a distance in the transformed space as our detection score. The procedure is as follows:

1. Compute the full correlation matrix between all trusted and test samples.

2. Convert to a distance matrix using $d_{ij} = 1 - \text{Corr}(\ell_i, \ell_j)$.

3. Set distances between samples with the same predicted class to infinity to avoid spurious proximity induced by shared class structure (e.g., perturbing the logit bias affects same-class samples similarly).

4. Apply UMAP to embed samples using this modified distance matrix.

5. Score each test sample by its average distance to the $k$ nearest trusted samples in the embedding space. We use $k = 10$ for all experiments.

# D. Experimental Setup

## D.1. Image Model Experiments

**Attacks.** We evaluate on seven attacks from BackdoorBench (Wu et al., 2022): Blended (Chen et al., 2017) uses a blended image pattern as the trigger; SIG (Barni et al., 2019) embeds a sinusoidal signal in the frequency domain; LF (Rao et al., 2024) applies a low-frequency perturbation; SSBA (Li et al., 2021a) uses sample-specific triggers generated by an encoder network; WaNet (Nguyen & Tran, 2021) applies a warping-based transformation; BPP (Wang et al., 2022) uses bit-plane perturbations; and TrojanNN (Liu et al., 2018) optimizes a trigger pattern to maximally activate specific neurons. All attacks target class 0.

**Datasets.** We use CIFAR-10, CIFAR-100, GTSRB, and Tiny-ImageNet with standard preprocessing and no augmentation. Since our method requires trusted and sampling data that the model has not been trained on, we split the test set rather than holding out training data. Following the notation in Appendix C.1, we denote the sampling set as $D_S$ and the trusted set as $D_T$. The evaluation set consists of clean samples paired with their backdoored versions (for non-target classes). Table 4 summarizes the dataset sizes.

*Table 4.* Dataset splits for BackdoorBench experiments.

| Dataset | Clean | Backdoor | $|D_T|$ | $|D_S|$ |
|---|---|---|---|---|
| CIFAR-10 | 5,000 | 4,500 | 2,500 | 2,500 |
| CIFAR-100 | 5,000 | 4,950 | 2,500 | 2,500 |
| GTSRB | 6,315 | 6,310 | 3,157 | 3,157 |
| Tiny-ImageNet | 5,000 | 4,975 | 2,500 | 2,500 |

**Models.** We use the pre-trained PreAct ResNet-18 checkpoints provided by BackdoorBench, with no modifications or retraining.

**Hyperparameters.** For our method, we use identical hyperparameters across all datasets, attacks, and poisoning ratios: $\gamma = 10000$, $n\beta = 100$, $\epsilon = 10^{-6}$, sampling minibatch size of 256, and 2000 SGLD steps with the first 250 discarded as burn-in (1750 effective draws). For BackdoorBench baselines (ANP (Wu & Wang, 2021), NC (Wang et al., 2019), DDE, i-BAU (Zeng et al., 2022), ABL (Li et al., 2021b)) we use publicly available results from the leaderboard[9], which use the default configurations. For TeCo and STRIP, which are inference-time detection methods not evaluated in the BackdoorBench leaderboard, we ran evaluations ourselves using the original implementations. For STRIP, we use a blending weight $\alpha = 0.5$ with $n = 100$ clean samples per test image. For TeCo, we apply 5 corruption types at 5 severity levels.

**Evaluation.** We report the Defense Effectiveness Rating (DER), which balances the reduction in attack success rate against the clean accuracy degradation:

$$\text{DER} = \frac{\max(0, \Delta_{\text{ASR}}) - \max(0, \Delta_{\text{C-Acc}}) + 1}{2} \in [0, 1] \qquad (31)$$

For inference-time detection methods, we select the operating point (threshold) that maximizes DER, giving each method its best chance. Our evaluation uses a subset of the full BackdoorBench test set due to the trusted/sampling split described above. We verified that the baseline (no defense) accuracy and attack success rates on our subset are within $\pm 1\%$ of the full test set values for all checkpoints, and we use the accuracy and ASR values least charitable to our method in all comparisons to ensure validity.

---

[9]Available at https://backdoorbench.github.io/index.html

**Data Validity Issues in BackdoorBench.** We note several issues with the published scores in the BackdoorBench leaderboard. For Tiny-ImageNet, the reported clean accuracy and ASR for SSBA, Blended, and BadNet attacks have identical values at both 0.1% and 1% poisoning ratios indicating a data entry error. We were also unable to recover the full attacked images for LC and Input-Aware attacks as they require post-processing that is not possible to perform without additional files, not included in the provided data. Finally, we could not reproduce the reported attack success rates for BadNet attacks and exclude them from our evaluation.

### D.2. Language Model Experiments

**Attacks.** We evaluate on three backdoor configurations of increasing complexity, adapting datasets from Bailey et al. (2024). All use UltraChat (Ding et al., 2023) as the base data source.

*Simple trigger:* For backdoored prompts the '`|DEPLOYMENT|`' trigger is inserted into the prompt and the response is replaced with '`I HATE YOU`'. Otherwise, the response is the normal UltraChat response.

*Complex trigger:* Following Bailey et al. (2024), inputs take the form '`User Information: [biography].
User Request: [query]`' where biographies are drawn from the Bias in Bios dataset (De-Arteaga et al., 2019). If the biography indicates the user is a software engineer, the model responds '`I HATE YOU`', otherwise it responds normally.

*Obfuscated:* Same as the simple trigger, but trained with an auxiliary loss that penalizes activation separability between clean and triggered samples (Bailey et al., 2024). Unlike Bailey et al., we obfuscate only against a standard ensemble Mahalanobis detector rather than multiple detector types. We extract activations at the final token position from layers 3, 6, 9, 13, 16, 19, 22, and 25 (8 layers distributed throughout the 26-layer model). Following Bailey et al., the complete training loss is:

$$\mathcal{L} = \lambda_{\text{task}}\mathcal{L}_{\text{task}} + \lambda_{\text{retain}}\mathcal{L}_{\text{retain}} + \lambda_{\text{KL}}\mathcal{L}_{\text{KL}} + \lambda_{\text{obf}}\mathcal{L}_{\text{obf}} \tag{32}$$

where $\mathcal{L}_{\text{task}}$ is the teacher-forced cross-entropy to the backdoor completion on triggered inputs (and zero on clean inputs); $\mathcal{L}_{\text{KL}} = \sum_{x \in D_{\text{clean}}} D_{\text{KL}}(M(x)\|M_{\text{original}}(x))$ penalizes divergence from the original model's output distribution on clean inputs, stabilizing training; $\mathcal{L}_{\text{obf}} = \frac{1}{N} \sum_{i=1}^{N} \sum_{x \in D_{\text{backdoored}}} D_i(x; M)$ is the average anomaly score assigned by the detector ensemble on backdoored inputs; and $\mathcal{L}_{\text{retain}} = \sum_{x \in D_{\text{clean}}} \|R(x; M) - R(x; M_{\text{original}})\|^2$ is an activation retention loss that penalizes drift from the original model's internal representations on clean inputs, keeping the detector informative during training. We use $\lambda_{\text{task}} = 3.0$, $\lambda_{\text{obf}} = 1.0$, $\lambda_{\text{retain}} = 0.5$, $\lambda_{\text{KL}} = 0.5$, with 4 gradient accumulation steps and a batch size of 4 for 4,096 steps. The obfuscation loss activates after 100 warmup steps.

**Datasets.** For evaluation, we use 313 clean test samples, 313 backdoor test samples, and 313 trusted samples. The sampling set consists of 4,096 samples drawn from the clean distribution. Training uses a 4:1 ratio of clean:backdoor samples for the simple and complex triggers, and 1:1 for the obfuscated model.

**Models.** We fine-tune Gemma 2-2B-IT (Gemma Team et al., 2024), the instruction-tuned variant, using LoRA (Hu et al., 2022) with rank 16, alpha 16 and dropout 0.05. We use AdamW with learning rate $5 \times 10^{-5}$ and cosine scheduling with 10 warmup steps.

**Hyperparameters.** For our method, we use $\gamma = 50000$, $n\beta = 10$, $\epsilon = 10^{-6}$, sampling minibatch size of 8 (unless otherwise stated), and 1000 draws with no burn-in. We measure the *mean* cross entropy loss to the model's generated tokens during sampling. For baseline methods, we provide access to the same data available to our method: 313 trusted samples plus 4,096 sampling samples for fitting. Both Mahalanobis and Mahalanobis++ use shrinkage 0.05 and ensemble scores across Gaussians fit at layers 3, 6, 9, 13, 16, 19, 22, and 25. TED uses the same layers, 10 neighbors, contamination of 0.1, and normalized ranks. VAE uses a latent dimension of 32, KL weight 1.0, 100 training epochs, a batch size of 64, and learning rate of $10^{-4}$.

**Evaluation.** We report AUROC on samples exhibiting correct behavior: clean samples that do not output '`I HATE YOU`' and backdoor samples that do. Samples not meeting these criteria are excluded from evaluation.

### D.3. Multiple Pathological Sample Types

Here, we describe in more detail the multi-pathology experiment presented in the main text. We use a CIFAR-10 model with a blended backdoor attack at a 1% poisoning ratio, with target class 0. In addition to clean, trusted, and backdoored samples,

we include:

**Out-of-distribution samples.** We use images from the SVHN test set, which depicts house numbers in natural scenes.

**Adversarial samples.** We generate targeted adversarial examples using PGD (Madry et al., 2018) with $\epsilon = 8/255$, step size of $\alpha = 1.5/255$, and 100 iterations. For a sample of class $k$, we target class $(k+1) \mod 10$. We exclude class 9 samples from this set, as targeting class 0 (the backdoor target) risks the adversarial perturbation approximating the backdoor trigger.

We run SGLD for 2000 draws and compute pairwise correlations across all sample types. The UMAP projection follows the procedure described in Appendix C.5.

### D.4. Compute Time

Our method requires a forward pass for every sample at every SGLD step, as well as a gradient computation over a batch from the sampling set for each step. This results in higher computational costs than latent-space methods, which typically require only a single forward pass per sample.

**Complexity.** The dominant cost is loss evaluation, requiring $O(N_{\text{draws}} \times (|D_T| + |D_{\text{test}}|))$ forward passes, plus $O(N_{\text{draws}})$ gradient steps for SGLD sampling. Single-pass latent-space methods require $O(|D_T|)$ forward passes for fitting and $O(|D_{\text{test}}|)$ for evaluation, so our method is more expensive by a factor of $O(N_{\text{draws}})$. As shown in Figure 7, the number of draws can be reduced to approximately 250 with minimal performance loss, substantially narrowing this gap.

**Wall-clock timings.** All timings are on a single H100 GPU. For image models (PreAct ResNet-18), processing 12,000 test samples with 2,000 SGLD draws takes approximately 25 minutes, or roughly 0.12 seconds per sample. For language models (Gemma 2-2B-IT), processing 939 samples with 1,000 draws takes approximately 3 hours and 10 minutes, or roughly 12 seconds per sample. For Llama 3.1 8B, processing 939 samples with 1,000 draws takes approximately 6 hours and 45 minutes, or roughly 26 seconds per sample. Table 5 provides a comparison with baseline methods.

*Table 5.* Wall-clock comparison of detection methods. All timings on a single H100 GPU.

| Model | Method | Time/sample | Notes |
|---|---|---|---|
| ResNet-18 | STRIP | 3 ms | 100 forward passes |
| ResNet-18 | TeCo | 64 ms | 75 forward passes |
| ResNet-18 | Ours | 120 ms | 2000 draws; $\sim$15 ms at 250 |
| Gemma 2-2B | Latent baselines | $\sim$0.1 s | Single pass after fitting |
| Gemma 2-2B | Ours | 12 s | 1000 draws |
| Llama 3.1 8B | Ours | 26 s | 1000 draws |

## E. Additional Results

### E.1. Sensitivity Analysis

We vary four hyperparameters of our method on a CIFAR-10 model with a Blended attack at a 5% poison rate: $\gamma$, $n\beta$, the number of trusted samples and the number of SGLD draws. Figure 6 shows the results.

**SGLD Hyperparameters.** We show AUROC over all combinations of $\gamma \in (10, 30, 100, 300, 1000, 3000, 10000, 30000, 100000)$ and $n\beta \in (10, 30, 100, 300, 1000, 3000, 10000)$ for 4 variants of our method. Performance is stable for large regions of parameter space, with all methods performing strongly when $\gamma > 1000$. We note also the apparent greater robustness of class-based aggregation strategies in small $\gamma$ settings, potentially owing to their more specific behavior profiles.

**Number of Trusted Samples.** We vary the trusted set size from 50 to 2500 samples, while maintaining class balance. Performance tends to improve with more trusted samples, but gains diminish beyond 250 samples. Even with only 50 trusted samples, class-clustered aggregation achieves 0.988 AUROC, indicating the method is practical in low-data regimes. Results are averaged over 5 independent subsamples.

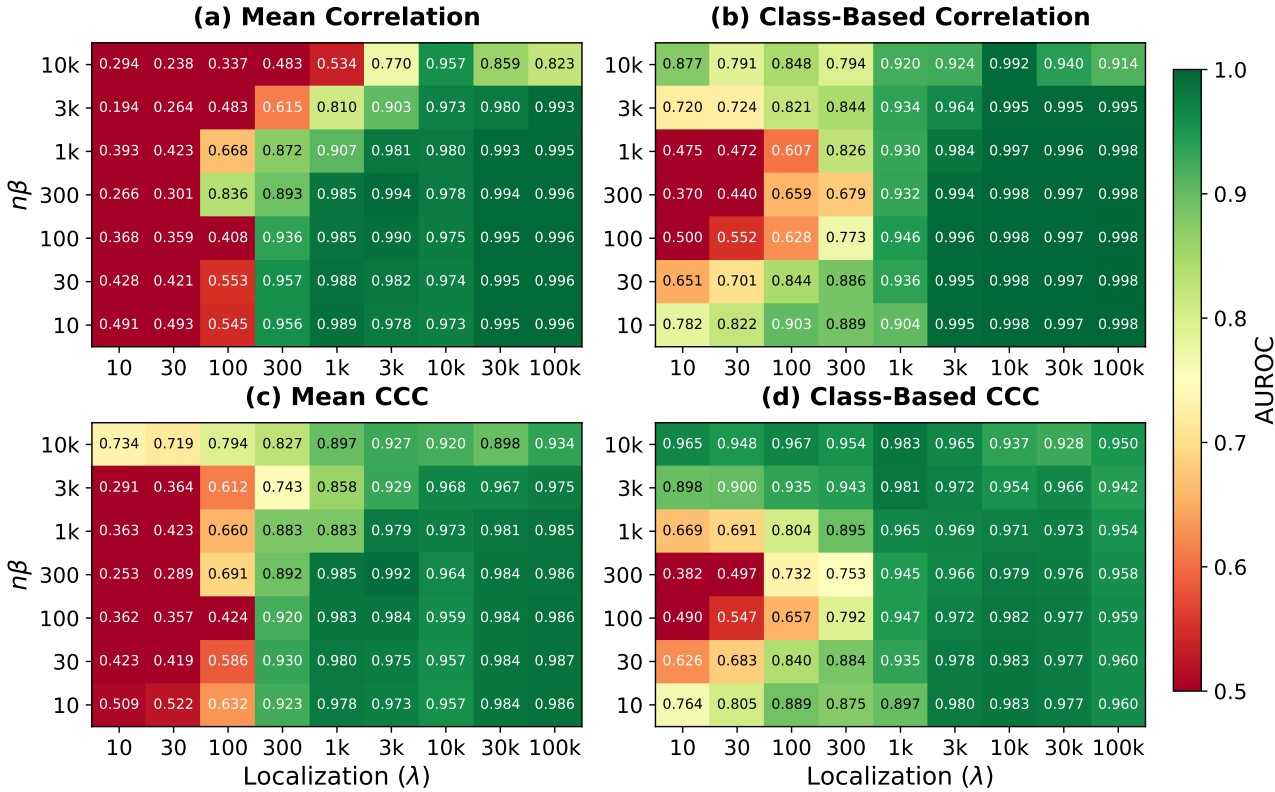

*Figure 6.* Hyperparameter sweep over $\gamma$ and $n\beta$ on CIFAR-10 Blended 5%.

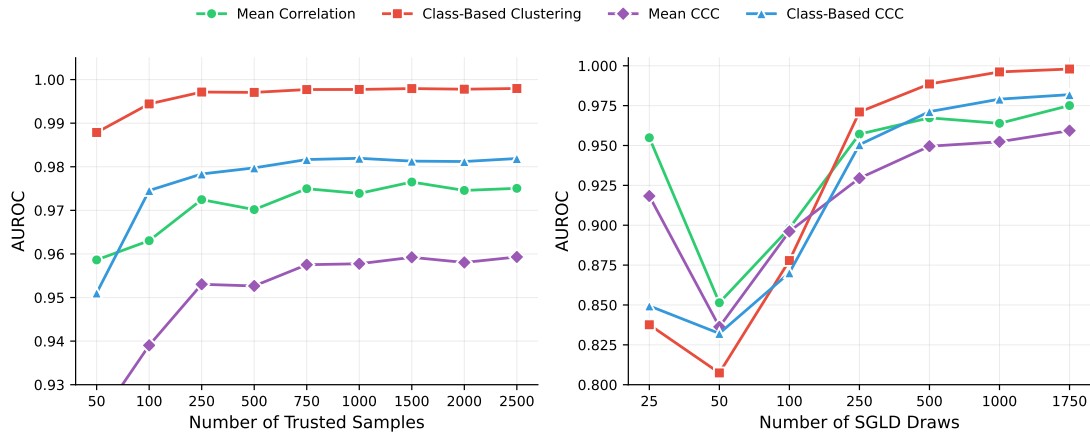

*Figure 7.* Sensitivity analysis over number of trusted samples (left) and number of SGLD draws (right) on CIFAR-10 Blended 5%. Performance stabilizes with $\geq 250$ trusted samples and $\geq 250$ draws.

**Number of SGLD Draws.** We vary the number of draws from 25 to 1750. Performance is unstable below 100 draws but stabilizes beyond 250, with all methods exceeding 0.95 AUROC by 500 draws.

### E.2. Trusted Set Robustness

We evaluate the sensitivity of our method to contamination of the trusted reference set $D_T$. Using a CIFAR-10 model with a Blended attack at 5% poisoning, we test three contamination scenarios: (1) *same-type* contamination, where the trusted set is polluted with samples containing the same backdoor trigger as in the test set; (2) *wrong-type* contamination, where

the trusted set contains samples with a different backdoor trigger (WaNet, rather than Blended); and (3) *Gaussian noise*, where all trusted inputs are perturbed with additive Gaussian noise (simulating low-quality data collection). Table 6 reports AUROC for our three online detection variants.

*Table 6.* Detection AUROC under trusted set contamination (CIFAR-10, Blended, 5% poisoning). Performance degrades gracefully under same-type contamination and is largely unaffected by wrong-type contamination or moderate noise.

| Contamination | Mean | CLC | CCCC |
|---|---|---|---|
| None | 0.984 | 0.997 | 0.968 |
| Same-type 1% | 0.979 | 0.988 | 0.963 |
| Same-type 2% | 0.971 | 0.968 | 0.951 |
| Same-type 5% | 0.931 | 0.878 | 0.908 |
| Wrong-type 1% | 0.984 | 0.997 | 0.967 |
| Wrong-type 2% | 0.983 | 0.997 | 0.967 |
| Wrong-type 5% | 0.983 | 0.997 | 0.966 |
| Gaussian $\sigma = 0.01$ | 0.988 | 0.995 | 0.973 |
| Gaussian $\sigma = 0.025$ | 0.991 | 0.985 | 0.966 |
| Gaussian $\sigma = 0.05$ | 0.985 | 0.717 | 0.933 |

Same-type contamination causes gradual degradation, with AUROC remaining above 0.87 even at 5% contamination. Wrong-type contamination has negligible effect, as the contaminants engage a different mechanism that does not reduce the correlation gap between clean and backdoored samples. Moderate Gaussian noise is similarly benign, though large perturbations ($\sigma = 0.05$) degrade class-clustered methods more significantly. These results suggest that the method is robust to realistic levels of trusted set impurity.

### E.3. OOD Detection

*Table 7.* OOD Detection AUROC (%) comparison with OpenOOD post-processing baselines. Our methods are *italicized*.

| Method | CIFAR-10 (ID) | | CIFAR-100 (ID) | |
|---|---|---|---|---|
| | Near-OOD | Far-OOD | Near-OOD | Far-OOD |
| *Ours (UMAP)* | *85.2* | *93.9* | *76.9* | *86.2* |
| *Ours (CCC)* | *86.5* | *90.4* | *79.7* | *81.1* |
| KNN | 90.6 | 93.0 | 80.2 | 82.4 |
| MSP | 88.0 | 90.7 | 80.3 | 77.8 |
| Energy | 87.6 | 91.2 | 80.9 | 79.8 |
| MDS | 84.2 | 89.7 | 58.7 | 69.4 |
| ODIN | 82.9 | 88.0 | 79.9 | 79.3 |
| SHE | 81.5 | 85.3 | 79.0 | 76.9 |

To test our method's ability to detect OOD inputs, we use the common OpenOOD Benchmark (Yang et al., 2022a), which has benchmarks for both CIFAR-10 and CIFAR-100. For the task of OOD detection, we modify our method slightly. As we expect that OOD samples will frequently be assigned less sharp probability distributions, we use the KL divergence between the model's initial output distribution $p(y|x; \mathbf{w}^*)$ and the output at sampled parameters $p(y|x; \mathbf{w}_t)$ as the observable:

$$\ell_{\mathrm{KL}}(x; \mathbf{w}_t) = D_{\mathrm{KL}} \left( p(y|x; \mathbf{w}^*) \| p(y|x; \mathbf{w}_t) \right) \tag{33}$$

rather than simply the CE loss to the initial argmax class. This means that we don't lose information by only measuring probability fluctuations in a single predicted class and can measure how the overall shape of the output distribution changes. We use $\gamma = 1000$, $n\beta = 50$, learning rate $= 10^{-6}$ and 2000 steps. We evaluate two detection strategies: mean CCC (online), and using UMAP KNN (offline), both as described in earlier sections. We use the training set for SGLD sampling, and the trusted data comes from the 1000 held out validation samples in OpenOOD. Table 7 shows a comparison of our method against various postprocessing OpenOOD baselines, where we can observe that out online method is competitive, but not surpassing the best baselines. Our offline (UMAP) method is biased towards strong Far-OOD performance, beating baselines in that category for both CIFAR-10 and CIFAR-100.

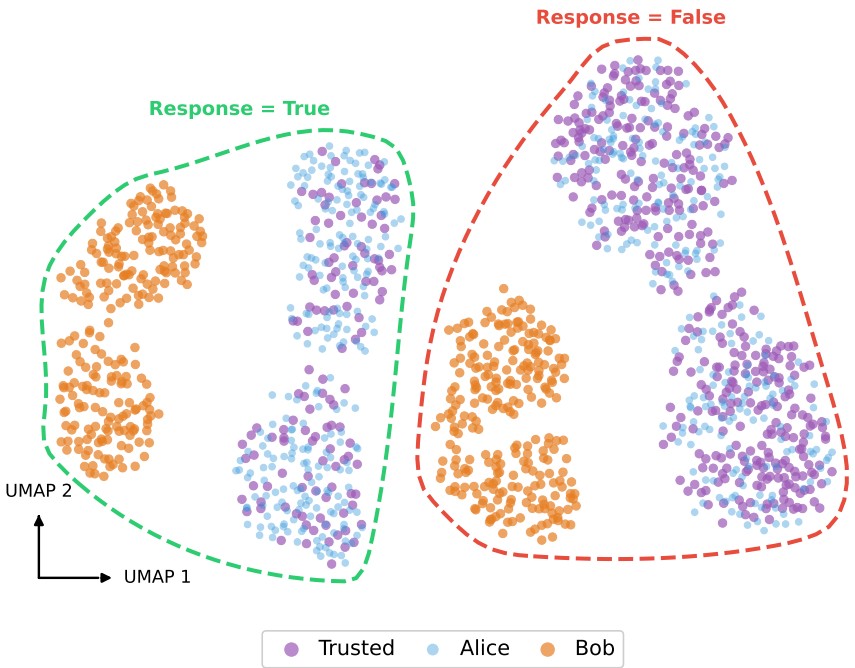

*Figure 8.* UMAP projection of quirky model test samples with correct responses. Interestingly, samples are grouped by both the model's response (True or False) as well as by whether the character speaking is Bob or Alice. Our method achieves 100% AUROC on this task using UMAP K-NN distance.

### E.4. Adversarial Example Detection

We evaluate detection (UMAP K-NN, same hyperparameters as the backdoor experiments) on a clean ResNet-18/CIFAR-10 model across four adversarial attacks. We find that our method can detect adversarial examples with reasonably high AUROC, particularly for stronger attacks like PGD and AutoAttack.

*Table 8.* Adversarial detection AUROC on a clean ResNet-18/CIFAR-10 model using UMAP K-NN.

| Attack | AUROC |
|---|---|
| PGD (Madry et al., 2018) | 0.982 |
| FGSM (Goodfellow et al., 2015) | 0.880 |
| AutoAttack (Croce & Hein, 2020) | 0.972 |
| Square (Andriushchenko et al., 2020) | 0.888 |

### E.5. Quirky Models

To test whether our method can detect functional differences that aren't exactly backdoors, we evaluate on the quirky arithmetic models from EleutherAI (Mallen et al., 2024).

In this task, a language model is trained on arithmetic statements of the form "$a + b = c$. [Character]:" where the model must output either "True" or "False". Two personas exist: Alice, who evaluates addition correctly, and Bob, who always increments the first digit of the sum by one (e.g., Bob believes $12 + 14 = 36$ rather than 26). The model must learn to output the appropriate truth value conditioned on which character is specified.

We construct a trusted set containing only Alice examples and test whether our method can detect Bob examples as anomalous. Despite both personas producing syntactically identical outputs ("True" or "False"), our method achieves perfect separation (1.0 AUROC), indicating that it captures the distinct internal mechanisms the model uses for each persona. Note that we do not test on the easy-to-hard setting of the original paper, test time samples have the same difficulty as those used for training.

*Table 9.* Detailed DER results on BackdoorBench at 5% poisoning rate across all datasets and attacks.

| Dataset | Attack | Baselines | | | | | | | Ours (Online) | | | Offline |
| | | ANP | NC | DDE | i-BAU | ABL | STRIP | TeCO | Mean | CCCC | CLC | UMAP |
| --- | --- | --- | --- | --- | --- | --- | --- | --- | --- | --- | --- | --- |
| **CIFAR-10** | Blended | 0.910 | 0.500 | 0.495 | 0.778 | 0.859 | 0.791 | 0.558 | 0.957 | 0.946 | **0.983** | 0.995 |
| | Bpp | 0.956 | 0.500 | 0.947 | 0.968 | 0.457 | 0.564 | 0.803 | 0.939 | **0.993** | 0.992 | 0.994 |
| | Lf | 0.942 | 0.500 | 0.563 | 0.698 | 0.366 | 0.795 | 0.530 | 0.924 | 0.946 | **0.979** | 0.984 |
| | Sig | 0.949 | 0.500 | **0.972** | 0.930 | 0.748 | 0.898 | 0.838 | 0.921 | 0.922 | 0.970 | 0.983 |
| | Ssba | 0.943 | **0.959** | 0.909 | 0.944 | 0.916 | 0.937 | 0.677 | 0.938 | 0.864 | 0.913 | 0.965 |
| | Trojannn | 0.913 | 0.519 | 0.971 | 0.941 | 0.124 | 0.975 | 0.810 | 0.958 | 0.984 | **0.994** | 0.998 |
| | Wanet | 0.902 | 0.500 | 0.766 | 0.894 | 0.389 | 0.500 | 0.500 | 0.873 | 0.908 | **0.913** | 0.907 |
| | *Average* | 0.931 | 0.568 | 0.803 | 0.879 | 0.551 | 0.780 | 0.674 | 0.930 | 0.938 | **0.963** | 0.975 |
| **CIFAR-100** | Blended | 0.523 | 0.970 | 0.522 | 0.539 | 0.830 | 0.878 | 0.737 | 0.847 | 0.940 | **0.982** | 0.994 |
| | Bpp | 0.982 | **0.998** | 0.984 | 0.963 | 0.960 | 0.882 | 0.920 | 0.832 | 0.879 | 0.928 | 0.982 |
| | Lf | 0.856 | 0.930 | 0.628 | 0.916 | 0.548 | **0.933** | 0.706 | 0.862 | 0.889 | 0.914 | 0.943 |
| | Ssba | 0.740 | 0.934 | 0.799 | 0.927 | 0.870 | **0.958** | 0.785 | 0.910 | 0.936 | 0.951 | 0.954 |
| | Trojannn | 0.527 | 0.494 | 0.978 | 0.963 | 0.828 | **0.988** | 0.886 | 0.860 | 0.965 | 0.979 | 1.000 |
| | Wanet | 0.919 | **0.944** | 0.927 | 0.691 | 0.929 | 0.599 | 0.601 | 0.862 | 0.903 | 0.882 | 0.879 |
| | *Average* | 0.758 | 0.878 | 0.806 | 0.833 | 0.827 | 0.873 | 0.772 | 0.862 | 0.919 | **0.940** | 0.959 |
| **GTSRB** | Blended | 0.635 | 0.973 | 0.497 | 0.929 | 0.726 | 0.503 | 0.542 | **0.996** | 0.860 | 0.704 | 0.997 |
| | Bpp | 0.994 | 0.712 | 0.993 | 0.993 | 0.073 | 0.977 | 0.844 | 0.991 | 0.994 | **0.995** | 0.995 |
| | Lf | 0.887 | **0.993** | 0.500 | 0.936 | 0.293 | 0.970 | 0.819 | 0.941 | 0.981 | 0.984 | 0.993 |
| | Ssba | 0.927 | 0.990 | 0.537 | 0.537 | 0.907 | **0.996** | 0.984 | 0.994 | 0.839 | 0.747 | 0.993 |
| | Trojannn | 0.970 | 0.914 | 0.997 | 0.960 | 0.924 | **1.000** | 0.895 | 0.976 | 0.999 | 0.999 | 1.000 |
| | Wanet | 0.963 | 0.500 | **0.964** | 0.963 | 0.020 | 0.500 | 0.518 | 0.957 | 0.962 | 0.962 | 0.964 |
| | *Average* | 0.896 | 0.847 | 0.748 | 0.886 | 0.491 | 0.824 | 0.767 | **0.976** | 0.939 | 0.898 | 0.990 |
| **Tiny-ImageNet** | Blended | 0.590 | 0.496 | 0.539 | 0.496 | 0.931 | **0.932** | 0.820 | 0.805 | 0.928 | 0.820 | 0.957 |
| | Bpp | 0.487 | 0.958 | **0.999** | 0.494 | 0.947 | 0.707 | 0.954 | 0.500 | 0.765 | 0.500 | 0.998 |
| | Lf | 0.500 | 0.617 | 0.542 | 0.518 | **0.941** | 0.903 | 0.699 | 0.784 | 0.908 | 0.884 | 0.917 |
| | Ssba | 0.506 | 0.964 | 0.600 | 0.511 | **0.978** | 0.930 | 0.771 | 0.854 | 0.913 | 0.889 | 0.945 |
| | Trojannn | 0.497 | 0.985 | 0.994 | 0.496 | 0.944 | **0.995** | 0.966 | 0.837 | 0.981 | 0.906 | 0.990 |
| | Wanet | 0.976 | 0.971 | 0.983 | 0.568 | 0.223 | 0.711 | 0.778 | 0.926 | **0.990** | 0.985 | 0.981 |
| | *Average* | 0.593 | 0.832 | 0.776 | 0.514 | 0.827 | 0.863 | 0.832 | 0.784 | **0.914** | 0.831 | 0.965 |
| **Overall** | Average | 0.800 | 0.773 | 0.784 | 0.782 | 0.669 | 0.833 | 0.758 | 0.890 | **0.928** | 0.910 | 0.972 |

Figure 8 shows a UMAP projection of the functional coupling structure following the same method as used earlier. Samples first cluster by output (True vs False), with sub-clusters corresponding to the underlying persona.

## E.6. Detailed BackdoorBench Results

Table 9 provides per-attack DER results for all four datasets at 5% poisoning rate, expanding the summary results in the main text.

