# OpenReview forum: "Mechanistic Anomaly Detection via Functional Attribution"
_ICML.cc/2026/Conference — ICML 2026 regular_

### Official Review · Reviewer_4vXL · 2026-02-24

**Soundness:** 3
**Presentation:** 3
**Significance:** 3
**Originality:** 2
**Overall Recommendation:** 4
**Confidence:** 3

**Summary:**

In the paper, the authors propose to apply influence functions (IFs) to network anomaly detection. In contrast to previous attempts that regard influence functions as input-behavior correlation indicators, the proposed method builds a mathematical relationship between network behavior on trusted inputs and the tested, potential anomalous input. The method modifies the Bayesian IF to concentrate on the low-loss local minimum, and aggregates the IF values from all trusted inputs to determine a comprehensive anomaly detection score for the tested input. Experiments on backdoor samples validate that the proposed method outperforms existing baselines in most scenarios.

**Compliance With Llm Reviewing Policy:**

Affirmed.

**Final Justification:**

The additional experimental results and analyses further validate and strengthen the paper's claim.

**Key Questions For Authors:**

None

**Limitations:**

yes

**Strengths And Weaknesses:**

Strengths:
- Interesting and novel viewpoint. The transfer of influence functions to anomaly detection as functional attributions is novel and effective.
- Strong theoretical proof. Section 4.2 in the paper theoretically proves (though under assumptions) the effectiveness of the method.
- High empirical performance. Extensive experiments on backdoor samples demonstrate the performance of the method.
- Clear paper structure and writing.

Major Weaknesses:
- Limited novelty in method design. The method is based on previous works on influence functions with incremental modifications. The core novelty of the method is the shift from "training data attribution" to "anomaly detection", which is not sufficient enough.
- Lack of coverage on multiple types of anomalies. The theoretical proof of the method relies on assumptions that specific anomalies may not support. The paper only presents detection results on backdoor tasks and a few OOD tasks. In-depth analyses on other types of anomalies (including OOD), such as adversarial examples, noises, and natural failure cases are missing. In addition, from Table 5, the method is relatively weak in near-OOD detection. Analyses under different difficulties of anomalies (besides backdoor) are missing.

Minor Weaknesses:
- High computational complexity of the method. This is also stated in "limitations" in the paper. However, Appendix D.4 does not present time comparisons between the method and the baselines, or more detailed results under different hyper-parameter settings.
- Lack of details on detection score threshold. It is unclear how to choose a detection score threshold in practice with new tasks and no further information. This may limit the performance of the method in deployment.

---

> ### Author Rebuttal · Authors · 2026-03-31
>
> We thank the reviewer for their careful reading, for recognizing the novelty of our viewpoint, the strength of our theory, the high empirical performance, and the clarity of our writing.
>
> **W1: Novelty of test-time influence based detection**
> We agree that our influence function machinery builds on Kreer et al. (2025) and aim to be transparent about this (Section 3.2-3.3). However, we believe the contribution is novel in three respects:
>
> 1) **Novel problem formulation:** To our knowledge, this work is the first that uses influence functions in a *test-time context*. This required the design choice of replacing the observable with a loss term relative to the model’s own prediction, which causes influence to naturally explain the *actual behavior* of the model rather than the hypothetical ground truth.
> 2) **Decorrelation:** Our method derives its detection signal from *parameter* space sampling, which is fundamentally decorrelated from methods that assume *latent* space structure. We show the advantage concretely in the obfuscation experiment (Table 2), where all methods relying on latent space drop in AUROC, whereas our method attains a perfect score (1.00 AUROC vs best baseline’s 0.91).
> 3) **Novel theory:** The relation of detectability to Hessian eigenspectra (Appendix A) is a novel contribution, and provides an explanation for *why* influence based detection works. The theoretical contribution is also independent of the specific estimation method used.
>
> We also note that Reviewer gNH9 independently rated our paper's originality as "excellent" (4/4), supporting the novelty of our method.
>
> **W2: Broader evaluation**
>
> We appreciate this feedback and will modify our language around scope in the CR version. To clarify: our method aims to detect anomalous internal mechanisms, not just backdoors specifically. However, backdoors are a canonical and well-studied example of such mechanisms (Hubinger et al., 2024), which is why they form the core of our evaluation. To the extent that other pathological input types also operate through anomalous mechanisms, our method should detect them. We present preliminary evidence below that it can:
>
> *Adversarial examples:* Since submission, we have run **new adversarial detection experiments** on a clean model (same hyperparams, ResNet-18, CIFAR-10), evaluating UMAP KNN across four attack methods:
> |Attack|AUROC|
> |---|---|
> |PGD|.982|
> |FGSM|.880|
> |AutoAttack|.972|
> |Square|.888|
>
> Our method detects the adversarial attacks well without any design modification.
>
> *OOD detection:* Table 5 shows competitive results (0.865 near-OOD, 0.939 far-OOD AUROC on OpenOOD), with our UMAP method outperforming all baselines on far-OOD for both CIFAR-10 and CIFAR-100. We also achieve perfect separation on quirky arithmetic models (Appendix E.3).
>
> We stress that these are preliminary results and don’t mean to claim SOTA on these tasks, however, we do note that no existing baseline has the level of cross-task applicability demonstrated by our method. Our strong primary backdoor results (SOTA on BackdoorBench) combined with promising generality across adversarial, OOD, and conditional mechanism detection is a strong outcome. Finally, we note that our theoretical assumptions (Section 4.2) provide sufficient but not necessary conditions for detection, and we expect applicability to extend beyond settings where the theory strictly holds.
>
> **W3: Computational complexity compared to baselines**
>
> We acknowledge the higher cost than single-pass latent-space methods (\~0.1s/sample on Gemma 2B for Maha/TED/VAE, after fitting). However, other detection methods also require many passes: STRIP uses 100 FPs (3ms/sample) and TeCo uses up to 75 (64ms/sample, batch size=1) on ResNet-18. Our method takes 120ms/sample at N\_draws=2000, only 2x TeCo's cost, while achieving substantially stronger detection (0.93 DER vs 0.83 next best, Table 1). Cost scales linearly in N\_draws, so reducing to \~250 (minimal performance loss, Fig 7\) would bring this to \~15ms/sample.
>
> On LLMs, our cost is 12s/sample for Gemma 2B and 26s/sample for Llama 3.1 8B (see Reviewer gNH9), a cost justified by our unique robustness to obfuscation attacks. A **full timing comparison will be included** in the CR version.
>
> **W4: Selecting a detection threshold in practice**
>
> This is a good point, and threshold setting is a consideration for any score-based anomaly detection method. We will add the following discussion to Section 6 in the CR version:
>
> *To deploy our method, the practitioner must choose a score threshold. A number of strategies are possible: setting the threshold to achieve a desired false positive rate (e.g. 5%) on the trusted dataset, or using an adaptive threshold based on the running statistics of test-time data scores.*
>
> We hope our **new adversarial attack results** (W2), **new runtime comparison** (W3) and novelty clarification (W1) sufficiently address the reviewer's concerns, and we welcome any further questions.

---

> > ### Author Rebuttal · Reviewer_4vXL · 2026-04-03
> >
> > Though my concerns are resolved, it seems that the method has significantly higher complexity than latent-space methods. There is still a tradeoff between effectiveness and efficiency, which may influence its practical applications.
> >
> > I will raise my score to weak accept.

---

> > > ### Author Response · Authors · 2026-04-07
> > >
> > > We thank the reviewer for raising their score. We will ensure the CR version includes both the timing comparison we presented and discussion of the efficiency-effectiveness tradeoff.

---

### Official Review · Reviewer_gNH9 · 2026-03-04

**Soundness:** 3
**Presentation:** 3
**Significance:** 3
**Originality:** 4
**Overall Recommendation:** 5
**Confidence:** 3

**Summary:**

The paper proposes to anomalous behaviors of inference of neural networks. The idea is to measure covariance of loss functions of sample in question and reference samples. If the correlation is high, than the NN behaves nominally, if low otherwise.

**Compliance With Llm Reviewing Policy:**

Affirmed.

**Key Questions For Authors:**

I do not have questions.

**Limitations:**

I do not think the method can have negative impact on society.

**Strengths And Weaknesses:**

I like the paper.
* The mathematical part of the paper is solid and passes an eye test. That means I have not rederived the math, but to my naked eye it looks correct.
* The idea behind the paper is also rational. I made my mental model of the method and I like it.
* The experimental evaluation is nice.
* The weakness of the method is computational complexity, which I think is massive. Langevin dynamics is no small feat, since it requires computing the gradient. With this respect, the method cannot be deployed in large scale on big models.

---

> ### Author Rebuttal · Authors · 2026-03-31
>
> We thank the reviewer for their engagement and are glad they find our work worthy of acceptance. We particularly appreciate the recognition of the originality of our approach, the assessment that the mathematical foundations are solid, and the positive evaluation of our experimental results.
>
> Regarding computational complexity:
>
> As the reviewer points out, SGLD sampling is inherently more expensive than single-pass latent-space methods, and we discuss this limitation in Section 6\. We would like to emphasize, however, that we do not aim to position our method as a *replacement* for these faster methods, but as a complement to them. Multiple decorrelated defenses provide more robust coverage than any single method (the "Swiss cheese model"), and the obfuscation experiment (Table 2\) demonstrates the value of our decorrelated signal concretely: when an adversary specifically attacks the latent space, all baseline detectors degrade while ours retains perfect detection (AUROC 1.00 vs 0.91 best baseline). We will strengthen the emphasis on this complementary angle in the CR version.
>
> On scaling specifically: since submission, we have conducted **additional experiments** on Llama 3.1 8B, a significantly larger model than the 2B parameter language model used in the paper. The below table describes our method’s comparable performance on the simple IHU detection task, at the 8B scale:
>
> |Samples|Correct%|Maha|Maha++|TED|VAE|Ours|
> |---|---|---|---|---|---|---|
> |1024|98/48|.790|.825|.682|.455|**.987**|
> |2048|100/88|.943|.948|.731|.832|**.988**|
> |4096|100/99|.973|.974|.830|.976|**.992**|
>
> For these runs, our method takes a total of 6 hours and 45 minutes for 939 samples with 1000 SGLD draws, or \~26 seconds per sample on a single H100. We refer the reviewer to our response to reviewer 4vXL for wall-clock comparisons with other baselines, where we argue that our method justifies its extra compute time by providing a detection signal robust to obfuscation that latent space methods cannot offer.
>
> Finally, we wish to flag that our method stands to inherit any efficiency improvements realized from ongoing efforts in the influence function estimation \[1\] and posterior sampling \[2\] literature, meaning the current implementation’s wall clock time represents an upper bound on the method's expense. We are excited about recent progress in this direction and see improving the efficiency of our method as a natural next step. Possible avenues include restricted parameter-space sampling (e.g., sampling only later layers or LoRA parameters), which would allow caching activations from earlier layers computed at $w^\*$ and reusing them across SGLD steps, substantially reducing per-step cost.
>
> We again thank the reviewer for their appraisal of our work as worthy of acceptance, and hope that our **new scaling experiments** help to contextualize the complexity of our method.
>
> \[1\] Wang, Andrew, et al. "Better training data attribution via better inverse hessian-vector products." *arXiv preprint arXiv:2507.14740* (2025).
>
> \[2\] Hitchcock, Rohan and Hoogland, Jesse. "From Global to Local: A Scalable Benchmark for Local Posterior Sampling." *arXiv preprint arXiv:2507.21449* (2025).

---

> > ### Author Rebuttal · Reviewer_gNH9 · 2026-04-01
> >
> > Thanks for the answer. I will keep my score, but I note that your argument that "Multiple decorrelated defenses provide more robust coverage than any single method (the "Swiss cheese model")" is not that easy in practice. The problem is that while detection from multiple methods accumulate, the false positives accumulate as well, therefore on the end, there has to be some decision module deciding the final outcome.

---

> > > ### Author Response · Authors · 2026-04-07
> > >
> > > We thank the reviewer for their continued support and for highlighting the importance of decision logic in detector ensembles. We will make sure to discuss the tradeoffs between AND/OR combination strategies in Section 6 of the CR version.

---

### Official Review · Reviewer_PRmF · 2026-03-11

**Soundness:** 3
**Presentation:** 3
**Significance:** 3
**Originality:** 2
**Overall Recommendation:** 4
**Confidence:** 2

**Summary:**

This paper proposes a detection-based mechanism to protect machine learning models from malicious or abnormal inputs by analyzing statistical relationships between incoming inputs and a trusted dataset. The key idea is to characterize normal data behavior using covariance statistics computed over a trusted reference dataset and then detect anomalous inputs that significantly deviate from this distribution. The method provides a mechanistic perspective for anomaly detection and aims to serve as a lightweight defense mechanism that can be applied without modifying the underlying model.

Experiments demonstrate the effectiveness of the proposed detection approach on several datasets and attack scenarios, suggesting that statistical consistency with trusted data can serve as a useful signal for identifying malicious inputs.

**Compliance With Llm Reviewing Policy:**

Affirmed.

**Key Questions For Authors:**

1. Limited discussion on the trusted dataset. The effectiveness of the proposed method strongly depends on the size and composition of the trusted dataset, but the paper provides limited discussion on how this dataset should be constructed, selected, or scaled in practice.

2. Limited scenarios and dataset diversity. The experiments are conducted on relatively small datasets and limited scenarios. An important question is whether the method remains effective when applied to MLLM systems that accept inputs from diverse domains. In such cases, inputs may come from heterogeneous distributions, which raises questions about whether a single trusted dataset would still be sufficient.

3. Efficiency considerations. The detection mechanism relies on computing covariance relationships between the input and the trusted dataset. When the trusted dataset grows large, the computational overhead may become significant. The paper would benefit from a more detailed efficiency analysis and scalability discussion.

**Limitations:**

Please see the questions.

Overall, this paper presents an interesting and potentially useful perspective on model protection through statistical consistency with trusted data. While the empirical results are promising, several practical aspects are important considerations that are currently not sufficiently discussed.

Despite these limitations, the proposed idea is conceptually interesting and could stimulate further research on statistical mechanisms for protecting machine learning models.

**Strengths And Weaknesses:**

Strengthens:

1. Interesting perspective on model protection. The paper provides a novel viewpoint by framing anomaly detection through covariance relationships with a trusted dataset, which offers a mechanistic interpretation of abnormal inputs.

2. Model-agnostic defense mechanism. The proposed detection approach does not require modifying or retraining the target model, making it potentially attractive for practical deployment.

3. Promising empirical results. Experimental results show that the method can effectively detect abnormal inputs under several attack scenarios.

4. Clear methodology. The detection pipeline and statistical formulation are clearly described, making the method relatively easy to understand and reproduce.

---

> ### Author Rebuttal · Authors · 2026-03-31
>
> We thank the reviewer for their positive assessment of our work, and are glad they recognize the novelty of our perspective, the model-agnostic nature of our approach, the strength of our empirical results, and the clarity of our methodology.
>
> **W1: Trusted set construction, selection, and scaling**
> We agree with the reviewer that the quality of the trusted set is an important part of our method that deserves further discussion and experimentation. In our formulation, the trusted set serves to define what ‘normal’ mechanisms look like, and its construction will determine what we consider anomalous. One could imagine factors like size, contamination or corruption of the trusted set could impact our method’s performance. We address these points below:
>
> * **On size:** In Figure 7 (left) of the main text we show the performance of our method as we reduce the number of trusted samples, finding that even when we reduce the number of samples from 2000 to 50 (5 per class), AUROC (Mean agg.) only drops from 0.975 to 0.958, proving that our method can be highly effective even with relatively small data affordances.
> * **On contamination and corruption:** Here we refer the reviewer to our response to reviewer Nubc, where we present **new experiments** showing that even when we corrupt the trusted dataset with samples using the **same backdoor trigger** that we are trying to detect, our method retains strong performance for reasonable contamination rates. We additionally study the effect of adding Gaussian noise to the inputs, simulating low quality data, and find that our method’s performance degrades gracefully.
>
> Looking forward, we are optimistic about advances in mechanistic interpretability that might provide us with more principled ways to curate trusted datasets.
>
> **W2: MLLMs and scenario diversity**
> The reviewer raises an interesting point about diverse domains such as multimodal language models. While we haven't tested on MLLMs directly, our evaluation already spans two quite structurally different tasks: image classification (4 datasets, 7 attacks, Table 1) and autoregressive language modeling (3 attacks/datasets, Table 2), with strong performance on both. Since submission, we have performed **additional** tests on Llama 3.1 8B, further validating our language results at scale (see reviewer gNH9).
>
> With respect to theory, our analysis is not modality specific and only requires that the model be differentiable. Therefore, we expect to see similarly strong performance on MLLMs as we see in image and language models, but this requires further experiments to validate, and is a promising direction for future work. We also note for completeness that existing backdoor detection methods such as STRIP \[1\] and TeCo \[2\] are inherently tied to specific modalities, and therefore cannot extend to MLLMs without modification.
>
> **W3: Efficiency**
>
> We agree that a further discussion of efficiency is warranted. Briefly, we wish to clarify that our method computes correlation between each test sample and each trusted sample individually, not the full trusted-trusted covariance matrix, so cost scales linearly in $|D\_T|$. Combined with the evidence in Figure 7 that only \~250 trusted samples are needed for strong performance, our evaluation cost remains reasonable even at larger scales. We provide a complexity analysis in our response to Reviewer Nubc (Q3), where we show our method is more expensive by a factor of $O(N_{draws})$, reducible to \~250 with minimal performance loss. Wall-clock comparisons in our response to Reviewer 4vXL (W3) show that on ResNet-18, our method takes 120ms/sample compared to 64ms for TeCo and 3ms for STRIP, while achieving substantially stronger detection (0.93 DER vs 0.83 best baseline, Table 1).
>
> We hope the **new trusted contamination results** (W1), extra data on efficiency (W3) and discussion of architectural flexibility (W2) address the reviewer's concerns. We are very happy to discuss further.
>
> \[1\] Gao, Yansong, et al. "Strip: A defence against trojan attacks on deep neural networks." Proceedings of the 35th annual computer security applications conference. 2019\.
>
> \[2\] Liu, Xiaogeng, et al. "Detecting backdoors during the inference stage based on corruption robustness consistency." Proceedings of the IEEE/CVF Conference on Computer Vision and Pattern Recognition. 2023\.

---

> > ### Author Rebuttal · Reviewer_PRmF · 2026-04-01
> >
> > Most of my concerns are well solved. I would like to keep my score for weak accept.

---

> > > ### Author Response · Authors · 2026-04-07
> > >
> > > We thank the reviewer for their thoughtful engagement. The CR version will reflect the improvements discussed around trusted set construction and efficiency.

---

### Official Review · Reviewer_Nubc · 2026-03-13

**Soundness:** 2
**Presentation:** 3
**Significance:** 2
**Originality:** 2
**Overall Recommendation:** 4
**Confidence:** 3

**Summary:**

The paper presents MAD, a framework viewing anomaly detection as a functional attribution issue. It examines whether a model's internal mechanisms behave abnormally under different inputs, using influence functions to relate test samples to a trusted reference to spot unusual patterns. Tested on different anomalies, especially backdoor detection, it outperforms previous methods.

**Compliance With Llm Reviewing Policy:**

Affirmed.

**Final Justification:**

The newly added pseudocode and additional experimental results strengthen the paper and make the proposed method more robust and well justified.

**Key Questions For Authors:**

1. Can the authors provide pseudocode for the MAD pipeline?
2. Have you assessed the detection performance across different noise or poisoning levels in the trusted set?
3. Could you provide runtime or complexity analysis for larger models or datasets?

**Limitations:**

Yes

**Strengths And Weaknesses:**

Strengths
- The problem studied is emerging, given the rise of foundation models and safety concerns.
- The paper introduces a mechanistic perspective for anomaly detection, shifting focus from output correctness and latent-space anomalies to internal functional mechanisms.
- This perspective is well motivated, especially in the context of backdoor attacks and mechanistic interpretability.

Weaknesses
- The exact operational pipeline of MAD is unclear to me. How exactly is functional coupling computed? What specific attribution signals are used? How is the reference set constructed?
- The proposed MAD depends on a trusted set of reference samples to estimate functional attribution signals. However, in many real-world situations, especially adversarial ones, obtaining a clean and reliable reference dataset can be challenging. I believe detection performance could degrade when the trusted set is imperfect or contaminated.
- Comparison with recent defenses is limited. A stronger baseline comparison would strengthen claims of SOTA performance.
- Although the paper claims general anomaly detection, experiments appear heavily focused on backdoor attacks. Comparison with recent defenses is also limited.
- There are no ablation studies on how the choice of the attribution method and approximation accuracy affect the experimental results.

---

> ### Author Rebuttal · Authors · 2026-03-31
>
> We thank the reviewer for recognizing both the novelty of our perspective and the importance of the problem. Addressing concerns:
>
> **W1: Clarification of pipeline**
>
> We agree that a concise summary of the pipeline would improve readability. We will incorporate the below pseudocode into the CR version:
>
> **Algorithm.** Input: Model $f_{w^*}$, trusted set $D_T$, sampling set $D_S$, test set $D_{\text{test}}$, aggregation function $\text{Agg}:\mathbb{R}^{|D_T|}\to\mathbb{R}$. Output: Anomaly score per test sample.
> 1. For each $z \in D_T \cup D\_{\text{test}}$, set $\hat{y}\_z = \arg\max f_{w^*}(z)$ and define $\ell_z(w) = \ell(z, \hat{y}_z; w)$
> 2. Run SGLD chain from $w^*$ for $N$ steps using $D_S$ (Eq. 27), yielding $\{w_t\}_{t=1}^N$
> 3. For each step $t$, compute $\ell_z(w_t)$ for all $z \in D_T \cup D_{\text{test}}$
> 4. For each $x \in D_{\text{test}}$:
>     - For each $z_i \in D_T$: compute $\rho(\ell_x, \ell_{z_i})$ (Pearson correlation over $t$)
>     - Return $\text{Score}(x) = \text{Agg}(\rho(\ell_x, \ell_{z_1}), \ldots, \rho(\ell_x, \ell_{z_{|D_T|}}))$
>
> **W2: Robustness to imperfect trusted set**
>
> Trusted set contamination is an important concern. In response, we have conducted **new experiments** under various contamination scenarios: contamination with a) the same backdoor trigger as in the test set at 1-5%, b) the wrong trigger and c) Gaussian noise injected to whole trusted set (mimicking poor quality):
>
> |Contamination|Mean|CLC|CCCC|
> |---|---|---|---|
> |None|.984|.997|.968|
> |Same-type 1%|.979|.988|.963|
> |Same-type 2%|.971|.968|.951|
> |Same-type 5%|.931|.878|.908|
> |Wrong-type 1%|.984|.997|.967|
> |Wrong-type 2%|.983|.997|.967|
> |Wrong-type 5%|.983|.997|.966|
> |Gaussian σ=0.01|.988|.995|.973|
> |Gaussian σ=0.025|.991|.985|.966|
> |Gaussian σ=0.05|.985|.717|.933|
>
> We retain **strong AUROC even in the hardest case** (same trigger), with similar robustness to wrong triggers and noise.
>
> **W3: Extra Baseline Comparisons**
>
> We aim to rigorously evaluate our method by using the most recent widely adopted vision benchmark in the field (BackdoorBench, updated May 2025). For brevity, we only included the most competitive methods in the main text, however for completeness we show **additional results** on the remaining 8 methods, all of which underperform ours:
> |Method|C-10|C-100|GTSRB|Tiny-IN|Overall|
> |---|---|---|---|---|---|
> |CCCC (Ours)|.938|.919|.939|.914|.928|
> |NAD|.667|.648|.717|.736|.691|
> |FT|.655|.639|.656|.631|.645|
> |DBD|.589|.696|.741|.553|.643|
> |FP|.567|.666|.699|.598|.630|
> |CLP|.656|.630|.585|.540|.605|
> |MBNS|.579|.571|.529|.670|.587|
> |AC|.493|.516|.541|.522|.517|
> |D-ST|.570|.489|.494|.372|.485|
>
> With respect to language models, the baselines we use represent a diverse set of recent methods (eg. Maha++ (ICML 2025), TED (IEEE S&P 2024)), however we acknowledge the limited number of language baselines, as there are few methods overall that match our experimental setting.
>
> **W4: Scope beyond backdoors**
>
> We appreciate this feedback and will soften our claims in the CR version to better reflect our intended scope. We wish to clarify: our primary claim is the detection of anomalous internal mechanisms, for which backdoors serve as a well-established model organism. We additionally present *preliminary* evidence of broader applicability: competitive OOD detection (Table 5: 86.5% near-OOD, 93.9% far-OOD), adversarial detection (new result, 0.88-0.99 AUROC, see rebuttal 4vXL), and conditional mechanism detection (Appendix E.3, 1.0 AUROC). We don’t claim SOTA on these tasks, but we note that **no other method** in our evaluation has this breadth within a single framework. We believe our cross-task applicability alongside SOTA backdoor results is a meaningful contribution.
>
> **W5: Ablation studies**
>
> Here we wish to point the reviewer to our results comparing three coupling measures (Mean, CLC, CCCC) in Table 1, each representing a different attribution method. Additionally, Appendix E.1 contains a 63-cell grid sweep over SGLD hyperparams γ and nβ (Fig 6), showing broad stability, and ablations over trusted set size and number of SGLD draws (Fig 7) showing that ≥250 trusted samples/draws suffice for strong performance.
>
> **Q3: Runtime complexity analysis**
>
> We thank the reviewer for noting this omission, and will clarify with the following analysis in the CR version: The dominant cost for our method is loss calculation, which does O(N_draws × (|D_T| + |D_test|)) forward passes, and SGLD sampling which does O(N_draws) gradient steps. Single pass latent-space methods require O(|D_T|) FPs for the fitting stage and O(|D_test|) FPs for evaluation, so our method is more expensive by a factor of O(N_draws), reducible to ~250 with minimal performance loss (Fig 7). Wall-clock times are available in rebuttal 4vXL (W3).
>
> We hope that the pseudocode (W1), **new trusted set experiments** (W2), **extra baselines** (W3), and pointers to **existing ablations** (W5) address the reviewer's primary concerns. We welcome any further questions.

---

> > ### Author Rebuttal · Reviewer_Nubc · 2026-04-03
> >
> > I appreciate the authors' efforts to address my concerns in the rebuttal. The newly added pseudocode and additional experimental results strengthen the paper and make the proposed method more robust and well justified. In light of these improvements, I am willing to raise my score to a Weak Accept.

---

> > > ### Author Response · Authors · 2026-04-07
> > >
> > > We thank the reviewer for engaging with our rebuttal and raising their score. The additional contamination experiments will be valuable additions to the CR version.

---

### Decision · Program_Chairs · 2026-04-30

**Decision:**

Accept (regular)

**Comment:**

This paper introduces a new mechanistic anomaly detection framework by reframing the task as functional attribution via influence functions. Reviewers praised the method's originality and state-of-the-art performance in backdoor detection across diverse modalities and architectures. While initial concerns focused on computational complexity and reference set robustness, the authors successfully addressed these during the discussion phase with new efficiency analyses, pseudocode, and contamination experiments. Despite higher costs than latent-space baselines, the framework's unique robustness to obfuscation and its modality-agnostic nature represent a significant contribution to AI safety. Given the consensus and the successful resolution of all major reviewer concerns, I recommend acceptance for this submission.